# Federated Variational Inference: Towards Improved Personalization and Generalization

**Elahe Vedadi**                                            *elahevedadi@google.com*
*Google Research*

**Joshua V. Dillon**                                            *jvdillon@google.com*
*Google Research*

**Philip Andrew Mansfield**                                            *memes@google.com*
*Google Research*

**Karan Singhal**                                            *karansinghal@google.com*
*Google Research*

**Arash Afkanpour**                                            *arashaf@google.com*
*Google Research*

**Warren Richard Morningstar**                                            *wmorning@google.com*
*Google Research*

**Reviewed on OpenReview:** *https://openreview.net/forum?id=6OmRkUHgs5*

## Abstract

Conventional federated learning algorithms train a single global model by leveraging all participating clients' data. However, due to heterogeneity in client generative distributions and predictive models, these approaches may not appropriately approximate the predictive process, converge to an optimal state, or generalize to new clients. We study personalization and generalization in stateless cross-device federated learning setups assuming heterogeneity in client data distributions and predictive models. We first propose a hierarchical generative model and formalize it using Bayesian Inference. We then approximate this process using Variational Inference to train our model efficiently. We call this algorithm *Federated Variational Inference (FedVI)*. We use PAC-Bayes analysis to provide generalization bounds for FedVI. We evaluate our model on FEMNIST and CIFAR-100 image classification and show that FedVI beats the state-of-the-art on both tasks.

## 1 Introduction

Federated Learning (FL) (McMahan et al., 2017) allows training machine learning models on decentralized datasets, avoiding the need to aggregate data on a central server due to privacy concerns. In FL, the central server oversees a global model distributed to clients who conduct local training, and the model updates are aggregated to iteratively improve the global model.

In simple and idealized settings, FL can approximate centralized training with similar theoretical guarantees, as seen in FedSGD (McMahan et al., 2017). However, real-world cross-device FL scenarios, such as those in (Reddi et al., 2020; Wang et al., 2021), often diverge from these ideal conditions. Practical FL implementations involve multiple local training steps to minimize communication overhead. Client participation is typically uneven, with some contributing more data and others not participating at all. Additionally, the non-Independently and Identically Distributed (non-IID) nature of client datasets, stemming from distinct data generation processes, challenges theoretical guarantees, leads to performance disparities between participating

and non-participating clients (Yuan et al., 2022), and complicates training high-performing models in practical FL setups.

Modern approaches address this challenge by either modifying the local loss to converge to a global solution (Li et al., 2020) or using personalized models to handle local distribution shifts (Zhang et al., 2022). Approaches for personalization have often focused on stateful FL setups, where clients are revisited throughout training and thus can update a locally stored model (Karimireddy et al., 2020; Wang et al., 2021). However, many production scenarios are effectively stateless, since individual clients only rarely contribute to training, and local models may be either stale or non-existent. Few studies have concentrated on personalization in this context. Those that have (Singhal et al., 2021), require clients to possess labeled examples for personalization.

This paper explores personalization in stateless cross-device FL setups and introduces Federated Variational Inference (FedVI), an algorithm which utilizes Variational Inference (VI) to enable models to generalize and personalize across diverse client data, even for untrained clients. The key contributions encompass (i) proposing a hierarchical generative model rooted in mixed effects models for cross-device federated setups, (ii) offering generalization bounds through Probably Approximately Correct (PAC)-Bayes analysis, (iii) introducing FedVI algorithm, inspired by the theoretical approach, which provides a simplified experimental approximation and can be implemented by the existing FL frameworks, and (iv) demonstrating the superior performance of FedVI on two federated datasets, FEMNIST and CIFAR-100, compared to previous state-of-the-art methods.

## 2 Related Work

**Bayesian FL:** To tackle statistical heterogeneity in FL, various studies have employed Bayesian methods to incorporate domain knowledge and aid convergence. Early attempts (Thorgeirsson & Gauterin, 2020; Chen & Chao, 2020) focused on model aggregation, either to retain uncertainty in model parameters, or to weight parameter updates proportional to performance. Zhang et al. (2022) instead attempts to use a Bayesian Neural Network (BNN) approximated with VI to train a global model using a Kullback–Leibler (KL) regularizer which induces convergence similar to the proximal term in FedProx (Li et al., 2020). While their local models can, in principle, personalize by deviating from the global model, they realistically require stateful settings with significant labeled data on clients in order to do so. Kotelevskii et al. (2022) casts personalized FL as mixed effects regression, and attempts to model the inherent heterogeneity in this setting explicitly using Stochastic Gradient Langevin Dynamics (Welling & Teh, 2011). Our proposed method assumes a similar generative process to Kotelevskii et al. (2022) but instead uses VI to efficiently infer the posterior, as well as place a bound on the predictive risk to induce generalization to new clients (Germain et al., 2016).

**Stateful FL:** There is a rich body of literature on personalization in cross device FL (Corinzia et al., 2019; Ghosh et al., 2020; Chen & Chao, 2021; Collins et al., 2021; Deng et al., 2020; Li et al., 2021; Hassan et al., 2023). Many previous methods focus on stateful settings, where local parameters are stored on clients and maintained across training rounds. However, as emphasized in Table 1 of (Kairouz et al., 2021), statelessness is a key characteristic of cross-device FL, highlighting its practical significance. Therefore, we focus on stateless settings, where maintaining up-to-date local states on each client is not feasible. This is similar to the setting considered by (Marfoq et al., 2022), who uses K-nearest neighbors to account for client distributional shift. While this is a robust means of dealing with both input and output distributional shift, it requires clients to possess labeled examples for every class (which is unrealistic in real-world setups), and cannot be used outside of classification problems. FedVI's statelessness stems from its use of an *amortized posterior inference model*. This concept shares similarities with how Variational Autoencoders (VAEs) (Kingma & Welling, 2013) perform inference. In VAEs, an encoder network maps input data to a latent space. FedVI takes a set of examples and infers the posterior over the local parameters, effectively capturing the personalized features of each client's data. This dynamic encoding allows for continuous adaptation and personalization as new clients join and contribute their data.

**Meta Learning:** There is a significant amount of prior work that studies connections between personalized FL and Model-Agnostic Meta-Learning (MAML) approaches (Finn et al., 2017; Singhal et al., 2021; Fallah et al., 2020; Collins et al., 2021; Lin et al., 2020; Chen et al., 2018). The main idea behind these works is to find an initial global shared model that the existing or new clients can adapt to their own dataset

by performing a few steps of gradient descent with respect to their local data. FedRecon (Singhal et al., 2021) is also motivated by MAML and considers a partially local federated learning setting, where only a subset of model parameters (known as global parameters) will be aggregated and trained globally for fast reconstruction of the local parameters. Our work can be considered as an extension of FedRecon. Unlike this work, we also provide a means of reconstructing local parameters [1] without access to labeled data.

## 3 Methods

### 3.1 Hierarchical Generative Model

Let us consider a stateless cross-device federated setup with multiple clients and a central server, where randomly selected client subsets participate in each training round. In this setup, we categorize each client's model parameters as global ($\theta$) and local ($\beta_k$ for $k \in [c]$[2]) parameters, with $c$ representing the total number of clients. Global parameters update at the server end after each training round, while local parameters are deleted after each round. Global parameters are drawn from the prior distribution $t(\Theta)$, while each client's local parameters are independent samples from the local prior $r(B_k)$. Additionally, data may not exhibit IID characteristics among clients, *i.e.*, $x_{ik} \sim \nu_k(X_k)$ for $i \in [n_k]$ and $k \in [c]$, where $n_k$ is the total number of data samples at client $k$. Moreover, each client may have a distinct predictive distribution. Although all clients share the same likelihood distribution family $\ell(Y_k|f(\theta, \beta_k, x_{ik}))$, the distribution varies based on $\beta_k$, making it different for each client.

The above setup is a prototypical example of a mixed effects model (Demidenko, 2013), commonly employed for predicting a continuous random variable using multiple independent factors, including both random and fixed, and incorporating repeated measurements from the same observational unit. Mixed effects models (Demidenko, 2013) have a well-established foundation. By framing our setup within this context, we can leverage existing theoretical insights in this field. To summarize, we propose the following hierarchical data generating process:

$$\theta \sim t(\Theta) \tag{1}$$
$$\text{for } k \in [c]:$$
$$\beta_k \sim r(B_k)$$
$$\text{for } i \in [n_k]:$$
$$x_{ik} \sim \nu_k(X_k)$$
$$y_{ik} \sim \ell(Y_k|f(\theta, \beta_k, x_{ik})),$$

where $f : \Phi \times \mathcal{B}_k \times \mathcal{X}_k \to \mathcal{Z}_k$ is a deterministic function (e.g., DNN) mapping what we know to the latent space $\mathcal{Z}_k$, which is the parameter space of our distribution over outcomes, $\ell(.)$.

For a more intuitive grasp of varying data generation processes and predictive distributions, consider the Federated EMNIST dataset (FEMNIST; Figure 1), where each client's dataset consists of numbers and letters handwritten by that client. Each client's input data reflects their unique writing style; for instance, a German client may include a horizontal middle bar when writing sevens, whereas an American client may not. Likewise, the German client may add a hood to the number 1, while the American client may not. This describes the difference in data generating distributions. This also illustrates that each client may have different predictive distributions: the American client may see the German's 1 as a 7, while the German client may see the American's 1 as a lowercase "l". Thus their predictive distributions are in direct conflict with each other. A purely global model cannot accommodate this diversity and must incorporate some level of local adjustments to accurately represent the data generation process. Our proposed algorithm explicitly assumes this data generating process. Note that this assumption reduces in special cases to existing FL setups, such as IID predictive distributions ($r(B_k) = \delta(B_k - \beta)$), or IID data generating processes ($\nu_k(X_k) = \nu(X_k)$). In the following section, we detail how we use VI to efficiently infer the model parameters.

---

[1] The detailed procedure for reconstructing the local parameters can be found in Section 5.
[2] In this paper we represent the set of $\{1, \ldots, c\}$ by $[c]$.

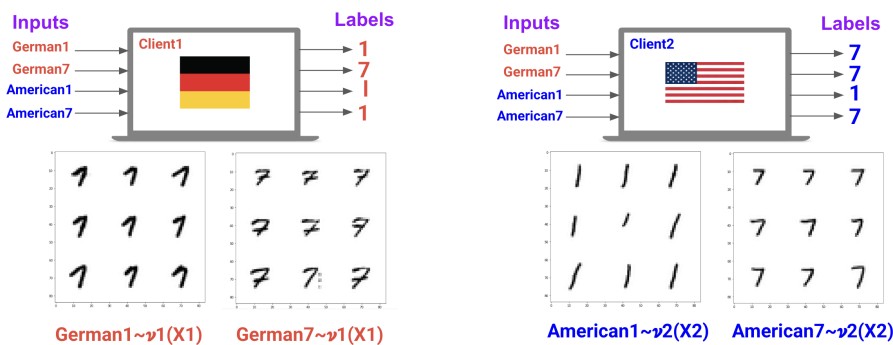

Figure 1: Illustration of diverse data generation and predictive models in cross-device FL.

## 3.2 Training Objective

In this section, our goal is to present a step-by-step definition of the objective function that is meant to be minimized throughout the training process. We begin by calculating the estimated probability density function of labels given input data, denoted as $\hat{p}(\{y^{n_k}\}^c) \stackrel{\text{def}}{=} p(\{y^{n_k}\}^c | \{x^{n_k}\}^c)$, following a similar marginalization approach as (Watanabe, 2018):

$$\hat{p}(\{y^{n_k}\}^c) \stackrel{\text{def}}{=} \int_\theta \int_{\beta_c} \cdots \int_{\beta_1} p(\theta, \beta^c | \{y^{n_k}, x^{n_k}\}^c) \ell(\{y^{n_k}\}^c | f(\theta, \{\beta_k, x^{n_k}\}^c)), \tag{2}$$

where $\beta^c \stackrel{\text{def}}{=} \{\beta_k\}^c \stackrel{\text{def}}{=} \{\beta_k : k \in [c]\}$, $x^{n_k} \stackrel{\text{def}}{=} \{x_i : i \in [n_k]\}$, $y^{n_k} \stackrel{\text{def}}{=} \{y_i : i \in [n_k]\}$, $\{x^{n_k}\}^c \stackrel{\text{def}}{=} \{x_{ik} : i \in [n_k], k \in [c]\}$, and $\{y^{n_k}\}^c \stackrel{\text{def}}{=} \{y_{ik} : i \in [n_k], k \in [c]\}$.

Therefore, for calculating $\hat{p}(\{y^{n_k}\}^c)$ it is required to calculate the posterior probability of model parameters given the training data which is equal to:

$$p(\theta, \beta^c | \{y^{n_k}, x^{n_k}\}^c) = \frac{p(\theta, \beta^c, \{y^{n_k}\}^c | \{x^{n_k}\}^c)}{p(\{y^{n_k}\}^c | \{x^{n_k}\}^c)}. \tag{3}$$

Assuming that the prior distribution of the global parameters, $t(\theta)$, the prior distribution of the local parameters, $r(\beta_k)$, and the likelihood distribution of each client, $\ell(y^{n_k} | f(\theta, \beta_k, x^{n_k}))$, are independent we calculate the numerator of Equation 3 as:

$$
\begin{aligned}
p(\theta, \beta^c, \{y^{n_k}\}^c | \{x^{n_k}\}^c) &= p(\theta, \{\beta_k, \{y_{ik}\}_{i \in [n_k]}\}_{k \in [c]} | \{x_{ik}\}_{k \in [c], i \in [n_k]}) \\
&= t(\theta) \prod_{k \in [c]} r(\beta_k) \prod_{k \in [c]} \prod_{i \in [n_k]} \ell(y_{ik} | f(\theta, \beta_k, x_{ik})) \\
&= t(\theta) \prod_{k \in [c]} \left( r(\beta_k) \prod_{i \in [n_k]} \ell(y_{ik} | f(\theta, \beta_k, x_{ik})) \right) \\
&= t(\theta) r(\beta^c) \ell(\{y^{n_k}\}^c | f(\theta, \beta^c, \{x^{n_k}\}^c)).
\end{aligned}
\tag{4}
$$

Moreover, the denominator of Equation 3 can be written as:

$$p(\{y^{n_k}\}^c | \{x^{n_k}\}^c) = \int_\theta \int_{\beta_c} \cdots \int_{\beta_1} p(\theta, \beta^c, \{y^{n_k}\}^c | \{x^{n_k}\}^c). \tag{5}$$

Unfortunately this integral is not only infeasible to compute, but also mathematically intractable. Consequently, this makes the whole posterior intractable.

To address the problem of the intractable posterior distribution, a tractable surrogate distribution, denoted as $q(\theta, \beta^c | \{y^{n_k}, x^{n_k}\}^c)$, is approximated using VI. By formulating a specific lower bound on the marginal distribution known as the evidence lower bound (ELBO), the best surrogate distribution can be obtained

by minimizing the ELBO (Equation 6). This minimization process provides the best approximation for the intractable posterior distribution, $p(\theta, \beta^c | \{y^{n_k}, x^{n_k}\}^c)$, under the chosen family of surrogates. The notation $D_{\mathrm{KL}}(q \| p)$ represents the KL divergence between two distributions $p$ and $q$, and detailed derivations of Equation 6 are available in Appendix A.

$$-\log p(\{y^{n_k}\}^c | \{x^{n_k}\}^c) \leq -\log p(\{y^{n_k}\}^c | \{x^{n_k}\}^c) + \min_q D_{\mathrm{KL}}(q(\theta, \beta^c | \{y^{n_k}, x^{n_k}\}^c) \| p(\theta, \beta^c, | \{y^{n_k}, x^{n_k}\}^c))$$

$$= \min_q \mathbb{E}_{q(\theta, \beta^c | \{x^{n_k}, y^{n_k}\}^c)} [\log \frac{q(\theta, \beta^c | \{x^{n_k}, y^{n_k}\}^c)}{p(\theta, \beta^c, \{y^{n_k}\}^c | \{x^{n_k}\}^c)}]. \tag{6}$$

By asserting factorization, we define the surrogate as a parametric distribution as:

$$q(\theta, \beta^c | \{y^{n_k}, x^{n_k}\}^c) \overset{\text{def}}{=} q_\lambda(\theta | \{y^{n_k}, x^{n_k}\}^c) \prod_{k \in [c]} q_\lambda(\beta_k | \theta, y^{n_k}, x^{n_k}) \overset{\text{def}}{=} q_\lambda(\theta | \{y^{n_k}, x^{n_k}\}^c) q_\lambda(\beta^c | \theta, \{y^{n_k}, x^{n_k}\}^c),$$
$$\tag{7}$$

where $\lambda$ is the parameter set that uniquely defines the surrogate distribution. Therefore, the objective function for training the proposed hierarchical model is a generalization of the negative ELBO (specifically, negative ELBO when $\gamma = \tau = 1$), which can be written as follows using the definition of KL divergence, logarithm properties, and the multiplication rule in probability.

$$\mathcal{J}(\lambda; \gamma, \tau) = \mathbb{E}_{q(\theta, \beta^c | \{x^{n_k}, y^{n_k}\}^c)} [\log \frac{q(\theta, \beta^c | \{x^{n_k}, y^{n_k}\}^c)}{p(\theta, \beta^c, \{y^{n_k}\}^c | \{x^{n_k}\}^c)}]$$

$$= \sum_{k \in [c]} \sum_{i \in [n_k]} \overbrace{\mathbb{E}_{q_\lambda(\theta | \{y^{n_k}, x^{n_k}\}^c) q_\lambda(\beta_k | \theta, y^{n_k}, x^{n_k})} [-\log \ell(y_{ik} | f(\theta, \beta_k, x_{ik}))]}^{\text{Per Datum Expected Loss}}$$

$$+ \underbrace{\gamma D_{\mathrm{KL}}(q_\lambda(\theta | \{y^{n_k}, x^{n_k}\}^c) \| t(\theta))}_{\text{Global Regularizer}} + \sum_{k \in [c]} \tau \underbrace{\mathbb{E}_{q_\lambda(\theta | \{y^{n_k}, x^{n_k}\}^c)} [D_{\mathrm{KL}}(q_\lambda(\beta_k | \theta, y^{n_k}, x^{n_k}) \| r(\beta_k))]}_{\text{Local Regularizer}}, \tag{8}$$

where $\gamma$, $\tau$, $t(\theta), r(\beta_k)$, and the functional form of $q_\lambda(\theta, \beta^c | \{y^{n_k}, x^{n_k}\}^c)$ are left as hyper parameters. The details of this derivation are provided in Appendix B. In the following section we explain how minimizing this objective function is equivalent to minimizing an upper bound on the generalization error.

## 4 Generalization Bounds

As mentioned earlier, we utilize a generalization of the negative ELBO as our objective function to train the hierarchical model. Minimizing this function ideally reduces the training dataset error (empirical risk). However, our primary aim is to minimize the error on unseen datasets (generalization error or true risk) for better generalization. To achieve this, we conduct a PAC-Bayes analysis, leveraging the results presented in Theorem 3 of (Germain et al., 2016). We introduce a slightly generalized version of this theorem in the form of the following corollary, enabling us to compute a generalization bound for the true risk of our model, under the assumption of non-IID empirical data samples.

**Corollary 1** *Given a distribution $\mathcal{D}$ over $\mathcal{X} \times \mathcal{Y}$, a hypothesis set $\mathcal{F} = \{\theta, \beta^c\}$, a loss function $\ell : \mathcal{F} \times \mathcal{X} \times \mathcal{Y} \to \mathbb{R}$, a prior distribution $\pi(\Theta, B^c) = t(\Theta)r(B^c)$ over $\mathcal{F}$, a $\delta \in (0, 1]$ and a real number $\eta > 0$, with probability at least $1 - \delta$ over the choice of $(\{x^{n_k}\}^c, \{y^{n_k}\}^c) \overset{\text{def}}{=} (X, Y) \sim \mathcal{D}$, for any $q(.)$ on $\mathcal{F}$ we have:*

$$\overbrace{\mathbb{E}_{\mathcal{D}}[-\log(\mathbb{E}_{q(\theta, \beta^c | X, Y)}[\ell(Y | X, \theta, \beta^c))]]}^{\text{True risk}} \leq$$

$$\overbrace{\mathbb{E}_{X,Y}[\mathbb{E}_{q(\theta, \beta^c | X, Y)}[-\log(\ell(Y | X, \theta, \beta^c))]]}^{\text{Empirical risk}} + \frac{1}{\eta} \Big[ \overbrace{D_{\mathrm{KL}}(q(\theta, \beta^c | X, Y) \| \pi(\theta, \beta^c))}^{\text{KL divergence}}$$

$$+ \underbrace{\log \left( \frac{1}{\delta} \mathbb{E}_{X,Y} \left[ \mathbb{E}_{\pi(\theta, \beta^c)} \left[ \exp \left( \eta \mathbb{E}_{\mathcal{D}}[-\log(\ell(Y | X, \theta, \beta^c))] - \eta \mathbb{E}_{X,Y}[-\log(\ell(Y | X, \theta, \beta^c))] \right) \right] \right] \right)}_{\text{Slack term}} \Big]. \tag{9}$$

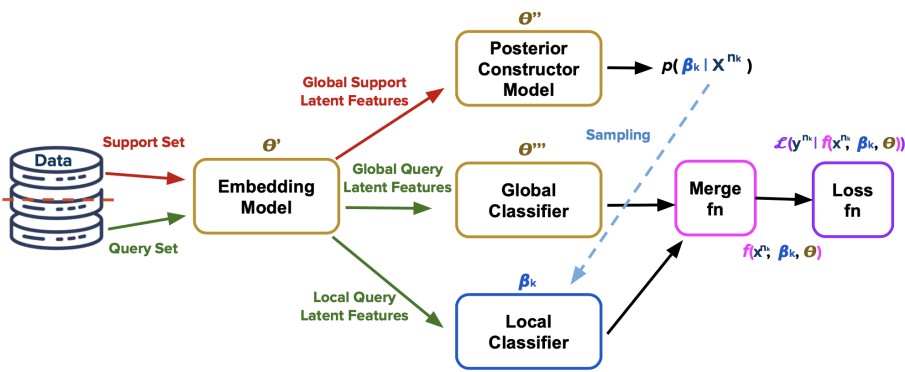

Figure 2: Our proposed model architecture implementing FedVI.

Where $\mathbb{E}_{X,Y}[\log(\ell(Y|X,\theta,\beta^c))] = \frac{1}{\frac{c}{\sum\limits_{k=1} n_k}} \sum\limits_{k=1}^{c} \sum\limits_{i=1}^{n_k} [\log(\ell(y_{ik}|x_{ik},\theta,\beta_k))]$ and $\mathbb{E}_{\mathcal{D}}[.] = \mathbb{E}_{(X,Y)\sim\mathcal{D}}[.]$.

**Sketch of Proof:** This corollary's proof closely follows Theorem 3 in Germain et al. (2016). We establish it using Jensen's inequality, Donsker-Varadhan change of measure inequality, and Markov's inequality. Additional details can be found in Appendix C.

Having obtained the generalization bound in Equation 9, we observe that it equals the negative ELBO (Equation 8) (for $\eta = 1$) plus a constant slack term, unrelated to the surrogate posterior distributions. Note that given Equation 9 validity for any $\eta > 0$, setting $\frac{1}{\eta} = \min\{\gamma, \tau\}$ allows us to consider Equation 8 plus a slack term as an upper bound on the True risk. Consequently, as long as this slack term remains finite, minimizing Equation 8 with respect to the surrogate distribution is equivalent to minimizing the generalization error with respect to the surrogate distribution. Thus we conclude that, assuming a finite slack term and with probability greater than $1 - \delta$, minimizing Equation 8 should improve the generalization of our model.

## 5 Implementation and Experimental Evaluation

The main goal of this section is to go through the details of our primary theoretical assumptions and model architecture for implementing an instance of our proposed hierarchical generative model and evaluating it. We note that the model architecture proposed in this section is one of infinity many architectures that is compatible with our theoretical model. This section will outline how the algorithm's steps correspond to the specific components of the hierarchical model and variational approximation presented in Section 3.

**Distributions:** For the prior distribution of the local parameters, we assume a normal distribution with zero mean and variance equal to that given by the initialization scheme (e.g. Glorot & Bengio, 2010; Glorot et al., 2011; He et al., 2015). We assume that clients' data generating distributions have the same support, but that they can be different from one another. We use a categorical distribution as our likelihood, where the logits generated by a deep neural network parameterized by $\theta$ and $\beta_k$ (described below). To simplify implementation, we use a point estimate for the global posterior. This is equivalent to assuming the hyper parameter of the global KL divergence is equal to zero, *i.e.,* in Equation 8 we have $\gamma = 0$. Moreover, to make sure that the KL divergence between the global posterior and the global prior, $D_{\mathrm{KL}}(q_\lambda(\theta|\{y^{n_k}, x^{n_k}\}^c)\|t(\theta))$, is finite we assume that the global posterior is a very narrow normal distribution, but still finite, while the global prior can be any finite function.

**Tasks:** We evaluate FedVI algorithm on two different datasets, FEMNIST[3] (Caldas et al., 2019) (62-class digit and character classification) and CIFAR-100[4] (Krizhevsky et al., 2009) (100-class classification). FEMNIST

---

[3]https://www.tensorflow.org/federated/api_docs/python/tff/simulation/datasets/emnist/load_data
[4]https://www.tensorflow.org/federated/api_docs/python/tff/simulation/datasets/cifar100/load_data

is particularly relevant since it has a naturally different data generative distribution for each client. Although CIFAR-100 data is synthetically partitioned using a hierarchical Latent Dirichlet Allocation (LDA) process (Li & McCallum, 2006) and distributed among clients, we evaluate FedVI on this dataset as well to show the superiority of our method on a more complicated classification task.

**Model Architecture:** There are infinitely many model architectures which could implement our method. The architecture that we chose in our experiments is illustrated in Figure 2 and summarised in Algorithm 1. The mathematical notations that are used in both Figure 2 and Algorithm 1 are as follows:

$$D_k \stackrel{\text{def}}{=} \{x^{n_k}, y^{n_k} : x^{n_k} \in \mathcal{X}_k,\ y^{n_k} \in \mathcal{Y}_k,\ k \in [c]\} \quad \text{(input dataset of client } k\text{)} \tag{10}$$

$$\mathcal{X}_k \stackrel{\text{def}}{=} \mathbb{R}^{i \times i \times j} \quad \text{(input space; whitened images)} \tag{11}$$

$$\mathcal{Y}_k \stackrel{\text{def}}{=} [\zeta] \quad \text{(label space)} \tag{12}$$

$$\mathbf{E}_{\theta'}(.) : \mathcal{X}_k \to \mathbb{R}^d \quad \text{(embedding model; relu-convnet with dropout)} \tag{13}$$

$$\mathbf{P}_{\theta''}(.) : \mathbb{R}^{d_g} \to \mathbb{R}^{(2 \cdot d_l + 1) \cdot |\mathcal{Y}_k|} \quad \text{(posterior constructor model; relu-mlp)} \tag{14}$$

$$\mathbf{G}_{\theta'''}(.) : \mathbb{R}^{d_g} \to \mathbb{R}^{|\mathcal{Y}_k|} \quad \text{(global classifier; one dense layer)} \tag{15}$$

$$\mathbf{L}_{\beta_k}(.) : \mathbb{R}^{d_l} \to \mathbb{R}^{|\mathcal{Y}_k|} \quad \text{(local classifier; one dense layer)} \tag{16}$$

$$\theta = \theta' \cup \theta'' \cup \theta''' \quad \text{(global parameters)} \tag{17}$$

$$\beta_k \quad \text{(local parameters of client } k\text{)}, \tag{18}$$

where for FEMNIST we have $i = 28$, $j = 1$, and $|\mathcal{Y}_k| = \zeta = 62$, and for CIFAR-100 $i = 32$, $j = 3$, and $|\mathcal{Y}_k| = \zeta = 100$, for $k \in [c]$. For both datasets the embedding size $d = 128$ and the number of local and global features are equal to $d_l = 26$ and $d_g = 102$, respectively.

Our proposed model architecture consists of four separate modules: an embedding model, $\mathbf{E}_{\theta'}(.)$, which encodes the input as a vector, a posterior reconstruction model, $\mathbf{P}_{\theta''}(.)$, which predicts the posterior over local parameters, a classifier parameterized by global parameters, $\mathbf{G}_{\theta'''}(.)$, and a classifier implemented by local parameters, $\mathbf{L}_{\beta_k}(.)$, generated by sampling from the reconstructed posterior. The global parameters serve the purpose of classifying input data samples by considering their global features shared among all clients. On the other hand, the local parameters play a distinct role in refining the classification outcome by accounting for the unique local features specific to each individual client. Our model follows the stateless definition outlined in Table 1 of (Kairouz et al., 2021), eliminating the necessity to retain prior client states for parameter updates. Clients are not required to store updated global parameters; instead, the server aggregates and transmits averaged updates for upcoming rounds. Furthermore, clients can avoid the need to store updated local parameters by employing the posterior constructor model in each round to reconstruct the local parameter distribution, allowing them to derive local parameters through sampling from this reconstructed posterior distribution.

**Implementation:** We implement our FedVI algorithm in TensorFlow Federated (TFF) and scale up the implementation to NVIDIA Tesla V100 GPUs for hyperparameter tuning. For FEMNIST dataset with 3400 clients we consider the first 20 clients as non-participating users which are held-out in training to better measure generalization as in (Yuan et al., 2022). At each round of training we select 100 clients uniformly at random without replacement, but with replacement across rounds. For CIFAR-100 with 500 training clients, we set the data of the first 10 clients as held-out data and select 50 clients uniformly at randomly at each round. We train FedVI algorithm on both FEMNIST and CIFAR-100 for 1500 rounds and at each round of training we divide both datasets into mini-batches of 256 data samples and used mini-batch gradient descent algorithm to optimize the objective function. The training procedure for each client $k$ at round $t$, outlined in Algorithm 1, is as follows. Further details regarding each step are explained subsequently:

1. Each client $k$ partitions its input data, $D_k$, over the batch dimension into support and query sets, $D_{k,s}$ and $D_{k,q}$, using the data split function, $S(.)$. Similar to FedRecon (Singhal et al., 2021), the support set is used to reconstruct the local parameters and the query set is used to make predictions.

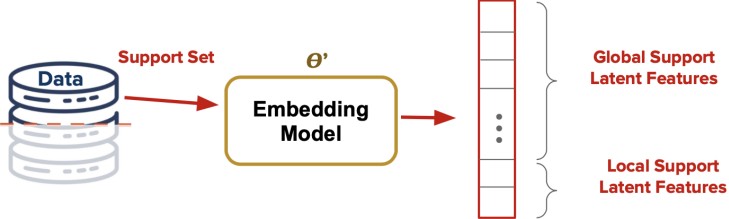

Figure 3: An illustration of the division of the data into support and query sets, as well as the division into global and local features.

Note that the support set we use can be unlabeled, and that the two sets need not be disjoint. However, we use disjoint sets in our experiments since (Singhal et al., 2021) found that it improved their model performance.

2. Both support and query sets are fed into the embedding model, $\mathbf{E}_{\theta'}(.)$, to extract vector representations of the data, *i.e.,* $R_{k,s} \in \mathbb{R}^d$ and $R_{k,q} \in \mathbb{R}^d$.

3. The representation for both support and query sets are further split over their features axis into global and local features, *i.e.,* $(R_{k,s}^g \in \mathbb{R}^{d_g}, R_{k,s}^l \in \mathbb{R}^{d_l})$ and $(R_{k,q}^g \in \mathbb{R}^{d_g}, R_{k,q}^l \in \mathbb{R}^{d_l})$, for $d = d_g + d_l$, using the feature split function $F(.)$, as illustrated in Figure 3.

4. The global features of the support set, $R_{k,s}^g \in \mathbb{R}^{d_g}$, are used to reconstruct the mean and variance of the local posterior, *i.e.,* $(\mu_k \in \mathbb{R}^{d_l \cdot |\mathcal{Y}_k|}, \sigma_k \in \mathbb{R}^{d_l \cdot |\mathcal{Y}_k|})$, through the posterior constructor model, $\mathbf{P}_{\theta''}(.)$. The local parameters, $\beta_k^{(t)}$, are generated by sampling from this posterior.

5. The global features of the query set, $R_{k,q}^g \in \mathbb{R}^{d_g}$, are passed to the global classifier, $\mathbf{G}_{\theta'''}(.)$, to get the global predictions, $O_k^g \in \mathbb{R}^{|\mathcal{Y}_k|}$, and the local features of the query set, $R_{k,q}^l \in \mathbb{R}^{d_l}$, and local parameters, $\beta_k^{(t)}$, are passed to the local classifier, $\mathbf{L}_{\beta_k}(.)$, to get the local modifications to the global predictions, $O_k^l \in \mathbb{R}^{|\mathcal{Y}_k|}$.

6. The local and global predictions are merged[5] to get the predictions. The log-likelihood is then computed between these predictions and labels and added to the KL divergence between local posterior and prior.

7. Global parameters get updated through back propagation over the loss function that is calculated in the previous step. This indirectly updates the local parameters, as they are inferred from the posterior constructor model, which is parameterized by the global parameters. Then the local update of the global parameters, $\Delta_k^{(t)}$, along with the number of query data samples at client $k$, $n_k$, are returned to the server.

8. The server aggregates all client updates and calculates the global update of the global parameters, $\theta^{(t+1)}$, and shares them with all clients $k \in \mathcal{S}^{(t+1)}$ for the next round of training.

**Data Partitioning:** First we note that for both FEMNIST and CIFAR-100 datasets, at each epoch we consider the first 50% of each mini-batch as the support set and the other 50% as the query set (*i.e,* for a mini-batch with 256 data samples the first 128 samples belong to the support set and the rest belong to query set). For the global-local features split, we found that using a larger number of global features (80%) than local features (20%) performed best. More specifically, in these experiments that the dimension of the last layer of the embedding model is equal to $d = 128$, the first 102 features are considered as the global features and the rest of 26 features are local features.

**Embedding Model:** In our experiments the embedding model, $\mathbf{E}_{\theta'}(.)$, is a relu convnet. For FEMNIST experiment we consider the convolutional model with 2 convolution layers that is described in Table 4 of

---

[5]While there are multiple ways that one could merge the predictions, we found that the simplest way was to add them together. This treats the local predictions as modifications to the global predictions in Logit space.

---

**Algorithm 1** FedVI Training

---

**Input:** set of global parameters $\theta$, data split function $S(.)$, feature split function $F(.)$, embedding model $\mathbf{E}_{\theta'}(.)$, posterior constructor model $\mathbf{P}_{\theta''}(.)$, global classifier $\mathbf{G}_{\theta'''}(.)$, local classifier $\mathbf{L}_{\beta_k}(.)$, merge function $f(.)$, client update algorithm $U(.)$.

**Server Executes:**
$\theta^{(0)} \leftarrow$ (initialize $\theta$)
**for** each round t **do**
$\quad \mathcal{S}^{(t)} \leftarrow$ (randomly sample $c$ clients)
$\quad$ **for** each client $k \in \mathcal{S}^{(t)}$ **in parallel do**
$\quad\quad (\Delta_k^{(t)}, n_k) \leftarrow \mathbf{ClientUpdate}(k, \theta^{(t)})$
$\quad$ **end for**
$\quad n = \sum_{k \in \mathcal{S}^{(t)}} n_k$
$\quad \theta^{(t+1)} \leftarrow \theta^{(t)} + \alpha_s \sum_{k \in \mathcal{S}^{(t)}} \frac{n_k}{n} \Delta_k^{(t)}$
**end for**

**ClientUpdate:**
$(D_{k,s}, D_{k,q}) \leftarrow S(D_k)$
$R_{k,s} \leftarrow \mathbf{E}_{\theta'}(x^{n_{k,s}}, \theta'^{(t)})$
$R_{k,q} \leftarrow \mathbf{E}_{\theta'}(x^{n_{k,q}}, \theta'^{(t)})$
$(R_{k,s}^g, R_{k,s}^l) \leftarrow F(R_{k,s})$
$(R_{k,q}^g, R_{k,q}^l) \leftarrow F(R_{k,q})$
$(\mu_k, \sigma_k) \leftarrow \mathbf{P}_{\theta''}(R_{k,s}^g, \theta''^{(t)})$
$\beta_k^{(t)} \leftarrow \text{sample}(\mathcal{N}(\mu_k, \sigma_k))$
$O_k^g \leftarrow \mathbf{G}_{\theta'''}(R_{k,q}^g, \theta'''^{(t)})$
$O_k^l \leftarrow \mathbf{L}_{\beta_k}(R_{k,q}^l, \beta_k^{(t)})$
$\theta_k^{(t)} \leftarrow U(f(O_k^g, O_k^l), y^{n_{k,q}})$
$\Delta_k^{(t)} \leftarrow \theta_k^{(t)} - \theta^{(t)}$
$n_k \leftarrow |D_{k,q}|$
return $(\Delta_k^{(t)}, n_k)$ to the server

---

(Reddi et al. (2020)) paper (without the top layer) and is parameteraized by the global parameters. the detailed structure of this embedding model is as the following.

For FEMNIST: $\mathbf{E}_{\theta'}(.) = conv(32) \rightarrow relu \rightarrow conv(64) \rightarrow relu \rightarrow maxpool(2,2) \rightarrow dropout(0.25) \rightarrow flatten \rightarrow dense(128) \rightarrow dropout(0.5)$

We choose a convolutional embedding model for CIFAR-100 as well, which is similar to FEMNIST embedding model, but having 5 convolution layers instead. The detailed structure is as follows.

For CIFAR-100: $\mathbf{E}_{\theta'}(.) = conv(32) \rightarrow relu \rightarrow conv(64) \rightarrow relu \rightarrow conv(128) \rightarrow relu \rightarrow conv(256) \rightarrow relu \rightarrow conv(512) \rightarrow relu \rightarrow maxpool(2,2) \rightarrow dropout(0.25) \rightarrow flatten \rightarrow dense(128) \rightarrow dropout(0.5)$

**Posterior Constructor Model:** The posterior constructor model, $\mathbf{P}_{\theta''}(.)$, is an MLP with three (dense) layers that takes the global features of the output of $\mathbf{E}_{\theta'}(.)$ as input and generates mean, variance, and bias of the posterior.

For both FEMNIST and CIFAR-100: $\mathbf{P}_{\theta''}(.) = dense(256) \rightarrow relu \rightarrow dense(256) \rightarrow relu \rightarrow dense((2 \times 26 + 1) \times |\mathcal{Y}_k|)$

**Global and Local Classifiers:** For both FEMNIST and CIFAR-100 experiments global classifier is one dense layer with $|\mathcal{Y}_k|$ units and no activation function, parameterized by the global parameters, and the local classifier is one dense layer similar to the global classifier, but parameterized by the local parameters.

**Optimizers:** We use Stochastic Gradient Descent (SGD) for our client optimizer and SGD with momentum for the server optimizer for all experiments (Reddi et al., 2020). We set the client learning rate equal to 0.03 for CIFAR-100 and 0.02 for FEMNIST dataset, and server learning rate equal to 3.0 with momentum 0.9 for both FEMNIST and CIFAR-100 datasets.

**Evaluation Results and Discussion:** We evaluate our proposed FedVI algorithm against state-of-the-art personalized FL method, KNN-Per (Marfoq et al., 2022), as well as FedPA (Al-Shedivat et al., 2020), FedEP (Guo et al., 2023) (using highest reported values), FedAvg+ (Chen & Chao, 2021), ClusteredFL (Ghosh et al., 2020), DITTO (Li et al., 2021), FedRep (Collins et al., 2021), APFL (Deng et al., 2020), and FedAvg (McMahan et al., 2017). Results for the baseline methods (except for FedPA and FedEP) are taken from (Marfoq et al., 2022).

Table 1: Test accuracy of the participating/non-participating clients. FedVI results are reported for $\tau = 10^{-9}$ for FEMNIST, and $\tau = 10^{-3}$ for CIFAR-100.

| Dataset | FedAvg | FedAvg+ | ClusteredFL | DITTO | FedRep | APFL | FedPA | FedEP | KNN-Per | **FedVI** |
|---|---|---|---|---|---|---|---|---|---|---|
| FEMNIST | 83.4/83.1 | 84.3/84.2 | 83.7/83.2 | 84.3/83.9 | 85.3/85.4 | 84.1/84.2 | 87.3/NA | 86.6/NA | 88.2/88.1 | **90.3/90.6** |
| CIFAR-100 | 47.4/47.1 | 51.4/50.8 | 47.2/47.1 | 52.0/52.1 | 53.2/53.5 | 51.7/49.1 | 46.3/NA | 50.7/NA | 55.0/56.1 | **59.1/58.7** |

KNN-Per (Marfoq et al., 2022) achieves personalized federated learning by combining a global MobileNet-V2 model with local k-nearest neighbors models based on shared data representations, demonstrating improved accuracy and fairness compared to other methods. FedPA (Al-Shedivat et al., 2020) reimagines federated learning as a posterior inference problem, proposing a novel algorithm that utilizes MCMC for efficient local inference and communication, employing CNN models for FEMNIST and ResNet-18 for CIFAR-100. FedEP (Guo et al., 2023) similarly reformulates federated learning as a variational inference problem, using an expectation propagation algorithm to refine approximations to the global posterior, and also employs CNN models for FEMNIST and ResNet-18 for CIFAR-100. APFL (Deng et al., 2020) offers a communication-efficient federated learning algorithm that adaptively combines local and global models to learn personalized models, utilizing various models (logistic regression, CNN, MLP) on diverse datasets, including FEMNIST and CIFAR-100. ClusteredFL (Ghosh et al., 2020) is designed for clustered users, iteratively estimating user clusters and optimizing their model parameters, demonstrating strong performance in various settings. Finally, FedRep (Collins et al., 2021) learns a shared representation and unique local heads for each client, using 5-layer CNNs for CIFAR and a 2-layer MLP for FEMNIST, while DITTO (Li et al., 2021) introduces a simple personalization mechanism within a multi-task learning framework, utilizing CNNs, logistic regression, and linear SVMs for different datasets.

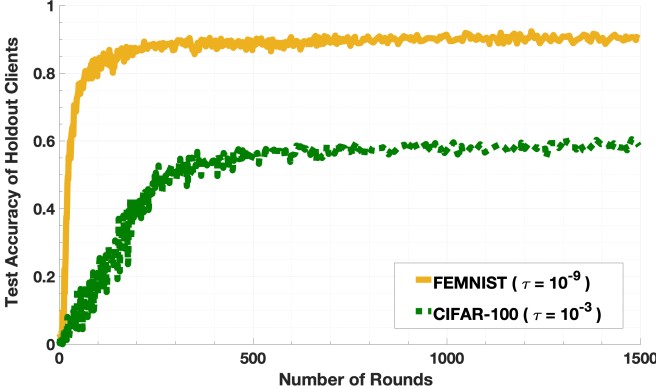

Figure 4: Non-participating test accuracy of FEMNIST ($\tau = 10^{-9}$) and CIFAR-100 ($\tau = 10^{-3}$) over 1500 rounds of training.

The performance of FedVI algorithm and other methods on the local test dataset of each client (unseen data at training) are provided in Table 1 for participating and non-participating (completely unseen during training) clients. All of the reported values are average weighted accuracy with weights proportional to local dataset sizes. To ensure the robustness of our reported results for FedVI, we average test accuracy across the last 100 rounds of training. Figure 4 illustrates the non-participating test accuracy on FEMNIST ($\tau = 10^{-9}$) and CIFAR-100 ($\tau = 10^{-3}$) over 1500 rounds of training, providing a visual representation of the results reported in Table 1.

Figure 5a shows the average test accuracy over the last 100 FEMNIST training rounds for a range of KL hyperparameter $\tau$, from 0 to 10. Notably, $\tau = 10^{-9}$ outperforms others, achieving higher accuracy with a smaller generalization gap compared to $\tau = 0$. Appendix D provides further insights and analysis regarding this experiment.

Figure 5b displays the average test accuracy over the last 100 rounds in CIFAR-100, with varying KL hyperparameter $\tau$. Notably, $\tau = 10^{-3}$ achieves the highest accuracy for both participating and non-

participating clients. Comparing $\tau = 0$ to other values ($\tau \neq 0$) reveals that minimizing KL divergence reduces the gap in participation test accuracy, as anticipated. Note that our objective function in Equation 8 is indeed a generalization of the ELBO, where setting $\tau = \gamma = 1$ recovers the standard ELBO. Setting $\tau = 0$ effectively removes the KL divergence term, resulting in maximum likelihood estimation (MLE). While MLE can sometimes lead to overfitting, the KL divergence term in the ELBO acts as a regularizer, promoting better generalization. Therefore, values of $\tau > 0$, which maintain the KL divergence term, generally exhibit superior performance and result is smaller generalization gaps. Furthermore, comparing this figure to Figure 5a, it's evident that the difference in test accuracy between $\tau = 0$ and $\tau = 10^{-9}$ in the FEMNIST experiment is significantly larger than the difference between $\tau = 0$ and $\tau = 10^{-3}$ in the CIFAR-100 experiment. This suggests that minimizing KL divergence is more critical for FEMNIST than for CIFAR-100. One possible explanation is that in FEMNIST, each client's data generation distribution naturally differs, while in CIFAR-100, data is synthetically partitioned and distributed among clients.

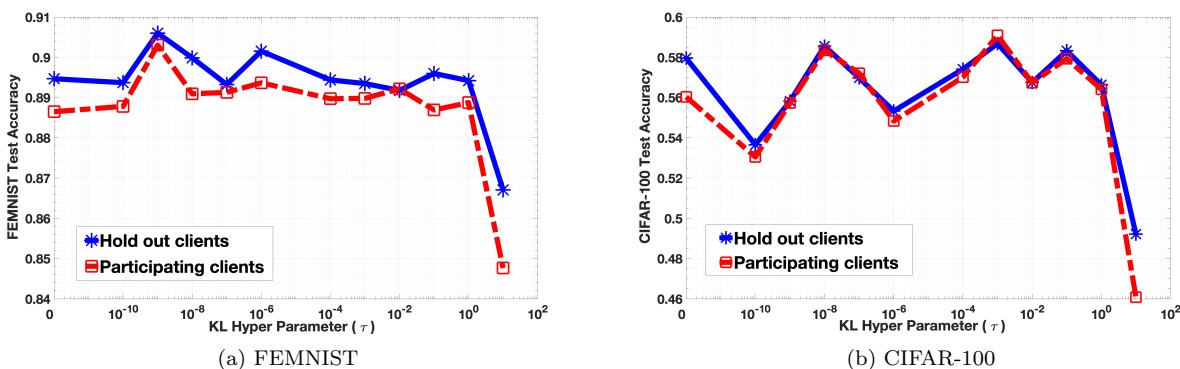

(a) FEMNIST

(b) CIFAR-100

Figure 5: Participating and non-participating test accuracy vs. KL hyperparameter $\tau$.

## 6 Conclusion and Future Work

This work addresses personalization in stateless cross-device federated setups through the introduction of FedVI, a novel algorithm grounded in mixed effects models and trained using VI. We establish generalization bounds for FedVI through PAC-Bayes analysis, present a novel architecture, and implement it. Evaluation on FEMNIST and CIFAR-100 datasets demonstrates that FedVI outperforms state-of-the-art methods in both cases. It is worth noting that in this paper, we employed a narrow normal distribution as the posterior for global parameters. However, in future research, we intend to explore more generalized distributions to enhance the modeling capabilities. Additionally, the model architecture presented in Figure 2 is just one of several possible architectures that align with our theoretical hierarchical model. In upcoming work we will focus on refining these architectures to optimize performance and explore their potential for achieving even better results.

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

# Appendices

## A    Derivations of Equation 6

Here we provide the detailed derivations of Equation 6 which are derived based on Section 2.2 of (Kingma & Welling, 2013). The main goal of these derivations is to devise an upper bound on the negative logarithm of the intractable denominator of the posterior probability of model parameters, *i.e.,* $p(\theta, \beta^c|\{y^{n_k}, x^{n_k}\}^c) = p(\theta, \beta^c, \{y^{n_k}\}^c|\{x^{n_k}\}^c)/p(\{y^{n_k}\}^c|\{x^{n_k}\}^c)$, to be able to approximate $p(\theta, \beta^c|\{y^{n_k}, x^{n_k}\}^c)$, in a tractable way. For this purpose, we consider an arbitrary distribution $q(\theta, \beta^c|\{y^{n_k}, x^{n_k}\}^c)$ as a surrogate for the posterior. Since the KL divergence of two distributions is always non-negative, we can use the KL divergence between the true posterior and our surrogate to devise an obvious and trivial upper bound on $-\log p(\{y^{n_k}\}^c|\{x^{n_k}\}^c)$ as the initial step in Equation 19. As the minimum of a non-negative number is always non-negative, we replace the KL divergence with its minimum value with respect to the surrogate distribution $q(\theta, \beta^c|\{y^{n_k}, x^{n_k}\}^c)$, to make this upper bound as tight as possible (Equation 20). Moreover, since $-\log p(\{y^{n_k}\}^c|\{x^{n_k}\}^c)$ is independent of the surrogate distribution, we move this term inside the minimum as shown in Equation 21. The rest of the proof comes from the definition of KL divergence, the multiplication rule of probability, and properties of logarithms. For the sake of simplicity in notation we have $\{y^{n_k}, x^{n_k}\}^c \stackrel{\text{def}}{=} X, Y$ in the following equations.

$$-\log p(Y|X) \leq -\log p(Y|X) + \overbrace{D_{\mathrm{KL}}(q(\theta, \beta^c|X, Y)\|p(\theta, \beta^c|X, Y))}^{\text{Always} \geq 0.} \tag{19}$$

$$\Rightarrow -\log p(Y|X) \leq -\log p(Y|X) + \overbrace{\min_q D_{\mathrm{KL}}(q(\theta, \beta^c|X, Y)\|p(\theta, \beta^c|X, Y))}^{\text{Always} \geq 0.} \tag{20}$$

$$\Rightarrow -\log p(Y|X) \leq \min_q -\log p(Y|X) + D_{\mathrm{KL}}(q(\theta, \beta^c|X, Y)\|p(\theta, \beta^c|X, Y)) \tag{21}$$

$$= \min_q \mathbb{E}_{q(\theta, \beta^c|X, Y)}[-\log p(Y|X) + \log \frac{q(\theta, \beta^c|X, Y)}{p(\theta, \beta^c|X, Y)}]$$

$$= \min_q \mathbb{E}_{q(\theta, \beta^c|X, Y)}[\log \frac{q(\theta, \beta^c|X, Y)}{p(\theta, \beta^c|X, Y)p(Y|X)}]$$

$$= \min_q \mathbb{E}_{q(\theta, \beta^c|X, Y)}[\log \frac{q(\theta, \beta^c|X, Y)}{p(\theta, \beta^c, Y|X)}] \tag{22}$$

## B    Derivations of Equation 8

We provide details for Equation 8, which is derived based on the definition of KL divergence, properties of logarithms, and the multiplication rule of probability. In the following equations $\{y^{n_k}, x^{n_k}\}^c \stackrel{\text{def}}{=} X, Y$ for the simplicity in notations.

$$p(Y|X) = \frac{p(\theta, \beta^c, Y|X)}{p(\theta, \beta^c|X, Y)} = \frac{p(\theta, \beta^c, Y|X)}{p(\theta, \beta^c|X, Y)} \times \frac{q(\theta, \beta^c|X, Y)}{q(\theta, \beta^c|X, Y)}$$

$$= \frac{p(\theta, \beta^c, Y|X)}{q(\theta, \beta^c|X, Y)} \times \frac{q(\theta, \beta^c|X, Y)}{p(\theta, \beta^c|X, Y)}$$

$$= \frac{t(\theta)r(\beta^c)\ell(Y|f(\theta, \beta^c, X))}{q_\lambda(\theta|X, Y)q_\lambda(\beta^c|\theta, X, Y)} \times \frac{q(\theta, \beta^c|X, Y)}{p(\theta, \beta^c|\theta, X, Y)}$$

$$\Rightarrow -\log(p(Y|X)) = -\log(\ell(Y|f(\theta, \beta^c, X)))$$

$$+ \log(\frac{q_\lambda(\theta|X, Y)}{t(\theta)}) + \log(\frac{q_\lambda(\beta^c|\theta, X, Y)}{r(\beta^c)}) - \log(\frac{q(\theta, \beta^c|X, Y)}{p(\theta, \beta^c|\theta, X, Y)})$$

$$\Rightarrow \mathbb{E}_{q(\theta,\beta^c|X,Y)}[-\log(p(Y|X))] = -\log(p(Y|X))$$

$$= \mathbb{E}_{q(\theta,\beta^c|X,Y)}[-\log(\ell(Y|f(\theta,\beta^c,X)))] + \mathbb{E}_{q(\theta,\beta^c|X,Y)}[\log(\frac{q_\lambda(\theta|X,Y)}{t(\theta)})]$$

$$+ \mathbb{E}_{q(\theta,\beta^c|X,Y)}[\log(\frac{q_\lambda(\beta^c|\theta,X,Y)}{r(\beta^c)})] - \mathbb{E}_{q(\theta,\beta^c|X,Y)}[\log(\frac{q(\theta,\beta^c|X,Y)}{p(\theta,\beta^c|X,Y)})]$$

$$\Rightarrow -\log(p(Y|X)) + D_{\mathrm{KL}}(q(\theta,\beta^c|X,Y)\|p(\theta,\beta^c|X,Y))$$

$$= \mathbb{E}_{q(\theta,\beta^c|X,Y)}[-\log(\ell(Y|f(\theta,\beta^c,X)))]$$

$$+ \mathbb{E}_{q_\lambda(\beta^c|\theta,X,Y)}[D_{\mathrm{KL}}(q_\lambda(\theta|X,Y)\|t(\theta))] + \mathbb{E}_{q_\lambda(\theta|X,Y)}[D_{\mathrm{KL}}(q_\lambda(\beta^c|\theta,X,Y)\|r(\beta^c))]$$

$$= \overbrace{\mathbb{E}_{q(\theta,\beta^c|X,Y)}[-\log(\ell(Y|f(\theta,\beta^c,X)))]}^{\text{Expected Loss}}$$

$$+ \underbrace{D_{\mathrm{KL}}(q_\lambda(\theta|X,Y)\|t(\theta))}_{\text{Global Regularizer}} + \underbrace{\mathbb{E}_{q_\lambda(\theta|X,Y)}[D_{\mathrm{KL}}(q_\lambda(\beta^c|\theta,X,Y)\|r(\beta^c))]}_{\text{Local Regularizer}}$$

$$= \mathbb{E}_{q(\theta,\beta^c|X,Y)}[-\log(\ell(Y|f(\theta,\beta^c,X)))] + D_{\mathrm{KL}}(q(\theta,\beta^c|X,Y)\|t(\theta)r(\beta^c)) \tag{23}$$

## C  Proof of Corollary 1

The proof of this corollary is derived from the proof of Theorem 3 in (Germain et al. (2016)). More specifically, Equation 24[6] comes from Jensen inequality, Equation 25 is a result of Donsker-Varadhan change of measure inequality, and Equation 26 comes from Markov's inequality.

$$\eta\mathbb{E}_{\mathcal{D}}[-\log\left(\ell(Y|X)\right)] = \eta\mathbb{E}_{\mathcal{D}}[-\log\left(\mathbb{E}_{q(\theta,\beta^c|X,Y)}[\ell(Y|X,\theta,\beta^c)]\right)]$$

$$\leq \eta\mathbb{E}_{\mathcal{D}}[\mathbb{E}_{q(\theta,\beta^c|X,Y)}[-\log\left(\ell(Y|X,\theta,\beta^c)\right)]] \tag{24}$$

$$\leq \eta\mathbb{E}_{X,Y}[\mathbb{E}_{q(\theta,\beta^c|X,Y)}[-\log\left(\ell(Y|X,\theta,\beta^c)\right)]]$$

$$+ D_{\mathrm{KL}}(q(\theta,\beta^c|X,Y)\|\pi(\theta,\beta^c))$$

$$+ \log\left(\mathbb{E}_{\pi(\theta,\beta^c)}[\exp\left(\eta\mathbb{E}_{\mathcal{D}}[-\log(\ell(Y|X,\theta,\beta^c))] - \eta\mathbb{E}_{X,Y}[-\log(\ell(Y|X,\theta,\beta^c))]\right)]\right) \tag{25}$$

$$w.p \stackrel{\leq}{>} 1-\delta \quad \eta\mathbb{E}_{X,Y}[\mathbb{E}_{q(\theta,\beta^c|X,Y)}[-\log\left(\ell(Y|X,\theta,\beta^c)\right)]] + D_{\mathrm{KL}}(q(\theta,\beta^c|X,Y)\|\pi(\theta,\beta^c))$$

$$+ \log\left(\tfrac{1}{\delta}\mathbb{E}_{X,Y}\mathbb{E}_{\pi(\theta,\beta^c)}\left[\exp\left(\eta\mathbb{E}_{\mathcal{D}}[-\log(\ell(Y|X,\theta,\beta^c))] - \eta\mathbb{E}_{X,Y}[-\log(\ell(Y|X,\theta,\beta^c))]\right)\right]\right) \tag{26}$$

We note that as opposed to Theorem 3 in (Germain et al., 2016), we did not assume the empirical data samples $(X,Y)$ are derived IID from a data distribution and interestingly this proof, which is a slightly revised version of the proof of Theorem 3 in (Germain et al., 2016), is correct for non-IID empirical data samples as well. The rationale behind this is that none of the steps in the aforementioned proof relies on the IID property of the empirical data samples. More specifically, this proof starts with calculating the true risk, $\mathbb{E}_{\mathcal{D}}$, and moving the logarithm inside the expected value using Jensen inequality. After that we use the Donsker-Varadhan inequality which says $\mathbb{E}_q[\phi(f)] < D_{\mathrm{KL}}(q\|\pi) + \log(\mathbb{E}_\pi[e^{\phi(f)}])$ (Germain et al., 2016). To use this inequality we define $\phi(f) = \mathbb{E}_{\mathcal{D}} - \mathbb{E}_{X,Y}$. The crucial aspect of this proof is the Donsker-Varadhan inequality, which holds true for any function $\phi(f) = \mathbb{E}_{\mathcal{D}} - \mathbb{E}_{X,Y}$ and whether the data we used to compute the empirical risk, $\mathbb{E}_{X,Y}$, is IID or not, doesn't affect its validity. Finally, the last inequality is the Markov's inequality that does not need IID assumption as well.

## D  Extended FEMNIST Experiment Analysis

In this appendix, we further analyze the FEMNIST experiment results to establish the robustness of the findings presented in Figure 5a. We demonstrate that local regularization consistently improves generalization

---

[6]Throughout this paper $\ell(Y|X)$ stands for marginalized likelihood over the surrogate posterior, therefore we have: $\ell(Y|X) = E_{q(\theta,\beta^c|X,Y)}[\ell(Y|X,\theta,\beta^c)]$.

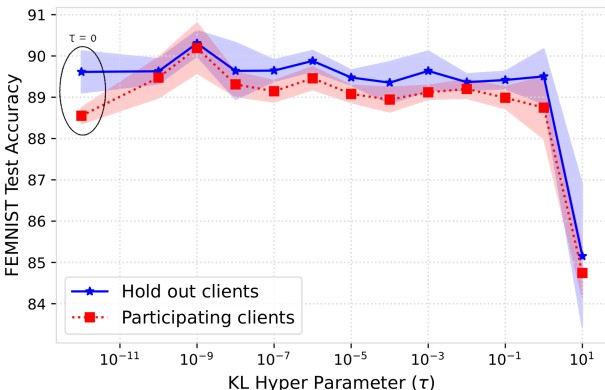

Figure 6: Average FEMNIST test accuracy for different kl hyper parameters ($\tau$), averaged across 5 runs with different random seeds.

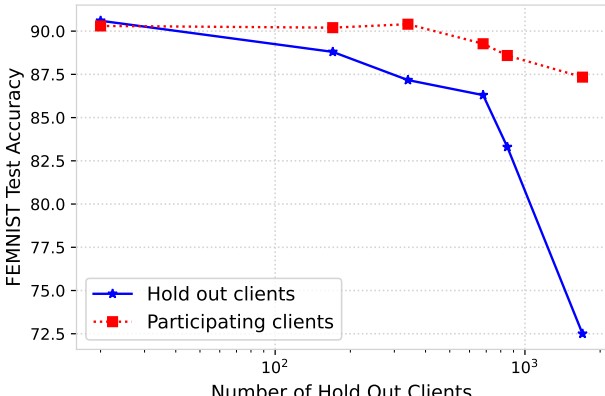

Figure 7: Test accuracy of FEMNIST ($\tau = 10^{-9}$) vs. number of hold out clients.

performance across different experimental setups, reducing the gap between non-participating and participating clients.

We conducted additional experiments, running the FEMNIST scenario five times with different seed values $(0, 10, 20, 30, 40)$. We then calculated the average and variance of both hold-out and participating test accuracy across these runs.

The results, presented in Figure 6, provide a smoother visualization of the trend. As the figure demonstrates, we still observe a noticeably larger generalization gap for $\tau = 0$ compare to $\tau > 0$. This supports our initial findings and further emphasizes the importance of the KL divergence term in promoting generalization, even with increased experimental rigor.

The experiment results provided in both Figure 5a and Figure 6 involved holding out only 20 clients (0.58% of the population) to ensure a fair comparison with other baselines. This small held-out set, might not fully represent the overall data distribution due to the inherent stochasticity in the remaining 3380 participating clients. To investigate this further, we conducted additional experiments on FEMNIST for $\tau = 10^{-9}$ with varying sizes of held-out sets (170, 340, 680, 850, and 1700 clients, representing 5%, 10%, 20%, 25%, and 50% of the population). As shown in Figure 7, these results indicate that increasing the size of the held-out set leads to an increase in the participation gap, with the participating clients eventually outperforming the nonparticipating ones. This suggests that the initial observation was indeed influenced by the small size of the held-out set and its potential deviation from the overall data distribution.

