# OpenReview forum: "Federated Variational Inference: Towards Improved Personalization and Generalization"
_TMLR — Accepted by TMLR_

### Review · Reviewer_BQTq · 2024-02-02

**Summary Of Contributions:**

This paper frames federated learning as a hierarchical probabilistic model, the key advantage of which being the ability to handle data heterogeneity. As inference is intractable, variational inference (VI) is proposed. The authors develop an approach in which point estimates of global parameters are maintained, and used to amortise inference over local parameters, the result being that local parameters do not have to be stored on individual clients. The method is evaluated on FEMNIST and CIFAR-100.

**Audience:**

Yes

**Broader Impact Concerns:**

N/A.

**Claims And Evidence:**

Yes

**Requested Changes:**

Please address the issues I have raised above, all of which I believe are critical to securing my recommendation for acceptance.

**Strengths And Weaknesses:**

It feels as though the paper is split into two distinct parts. In the first, the authors motivate the use of a hierarchical probabilistic model to handle data heterogeneity across clients, and VI for handling the challenge of inference. The generalisation bounds section is a nice addition; however, my main concern is in the correct handling of the hyper parameters $\gamma$ and $\tau$. For anything other than $\gamma, \tau = 1$, (8) does not lower-bound the evidence, so is not an ELBO as described. The authors should make this clear. Further, I suspect that Corollary 1 is only valid for $\gamma$ = $\tau$ also? I could be incorrect on this—clarification from the authors would be appreciated. In the second section, the authors describe their FedVI algorithm. Although one can piece together the parts of this algorithm to relate it back to the hierarchical model and variational approximation described in Section 3, this connection is far from explicit.

Some examples of things that need to be made clearer:

(1) the likelihood is parameterised by combining the outputs of two different functions, one using global and the other using local parameters. (a) make this clear and (b) how does this combination happen?.

(2) The approximate posterior over local parameters, which itself should be clearly defined, is amortised given a subset of the observed data (it would be nice for connections to be drawn to other methods that use amortised inference, such as VAEs). (a) If this is what makes it “stateless”, then this should be made clear. (b) Also, if this is the key contribution, which I think it is, then highlight it. (c) I’m not convinced that this is actually doing VI (i.e. targeting the ELBO), as the expectation in (8) is now over the approximate posterior given the support set and the likelihood is only evaluated at a non overlapping query set. This feels closer to the neural process ML objective (see e.g. [1]). (3) How the posterior constructor model operates as a set function on the set of global features of the query set. Something that would help here is clearly defining the space in which the intermediate variables (e.g. $R^g_{k, q}$) exist.

(3) I understand the notion of “stateless” to mean “no local parameters”. It would be great as to why this is beneficial in this case. Citations to use cases in the introduction would help, and clarity on why it’s an issue to store local parameters vs. computing them using a global model (aren’t these then stored locally anyway to obtain the local predictions?). I’m also somewhat confused by “local parameters remain on clients” in section 3.1—isn’t this exactly what you’re trying to avoid by being stateless?

(4) There is no description of the baseline methods. Are the architectures used the same as the global model? At the bare minimum these should be included in the appendix.

(5) “No assumptions are made about clients’ data generating distributions”. I’m not sure what you mean by this, as you’re using a very specific CNN architecture and categorical distribution which indicates that you are making assumptions.


[1] Foong et al. (2020): Meta-learning stationary stochastic process prediction with convolutional neural processes, section 3.2.

---

> ### Author Response · Authors · 2024-04-25
> **Review Responses**
>
> We would like to thank the reviewer for the constructive comments, which have really helped us to improve the paper. In what follows we replied to the reviewer questions point by point.
>
> **It feels as though the paper is split into two distinct parts. In the first, the authors motivate the use of a hierarchical probabilistic model to handle data heterogeneity across clients, and VI for handling the challenge of inference. The generalisation bounds section is a nice addition; however, my main concern is in the correct handling of the hyper parameters $\gamma$ and $\tau$. For anything other than $\gamma$, $\tau$ = 1 (8) does not lower-bound the evidence, so is not an ELBO as described. The authors should make this clear. Further, I suspect that Corollary 1 is only valid for $\gamma = \tau$ also? I could be incorrect on this—clarification from the authors would be appreciated.**
>
> We appreciate the reviewer's insightful question. While Equation (8) is indeed a generalization of ELBO (specifically, ELBO when $\gamma=\tau=1$), Corollary 1 generalizes this bound to settings where $\gamma, \tau \neq 1$. Note that while Equation (8) may not be specifically a lower bound on the evidence, it is an upper bound on the True risk which is equally useful in practice. More specifically given Equation (9)'s validity for any $\frac{1}{\eta} >0$, setting $\frac{1}{\eta} = \min(\gamma, \tau)$ makes Equation (8) a valid upper bound on the True risk. For the mathematical proof please check our response to this question in the uploaded file.
>
> **In the second section, the authors describe their FedVI algorithm. Although one can piece together the parts of this algorithm to relate it back to the hierarchical model and variational approximation described in Section 3, this connection is far from explicit.**
>
> We appreciate the reviewer's feedback regarding the clarity of the connection between the FedVI algorithm (Section 5) and the hierarchical model/variational approximation (Section 3). We acknowledge that this link could be made more explicit. To address this, we have added the following introductory paragraph at the beginning of Section 5, directly relating the algorithm to the theoretical framework:
> *The main goal of this section is to go through the details of our primary theoretical assumptions and model architecture for implementing an instance of our proposed hierarchical generative model and evaluating it. We note that the model architecture proposed in this section is one of infinity many architectures that is compatible with our theoretical model.  This section will outline how the algorithm's steps correspond to the specific components of the hierarchical model and variational approximation presented in Section 3.*
>
> **1. The likelihood is parameterised by combining the outputs of two different functions, one using global and the other using local parameters. (a) make this clear and (b) how does this combination happen?**
>
> We thank the reviewer for this suggestion.  While there are multiple ways that one could merge the predictions, we found that the simplest way was to add them together.  This treats the local predictions as modifications to the global predictions in Logit space. To improve clarity for the reviewer, we have added an explanation as a footnote on page 8.
>
> **2. The approximate posterior over local parameters, which itself should be clearly defined, is amortised given a subset of the observed data (it would be nice for connections to be drawn to other methods that use amortised inference, such as VAEs). (a) If this is what makes it “stateless”, then this should be made clear. (b) Also, if this is the key contribution, which I think it is, then highlight it.**
>
> The reviewer's suggestion is greatly appreciated. We agree that the connection to other amortized models such as VAEs should be made clear and have mentioned this connection in the paper.  This is indeed the element that makes the model "stateless", since performing this inference during training/evaluation removes the need to maintain a set of local parameters for each client.  We had attempted to highlight this benefit of our inference procedure in several places throughout the paper, but had fallen short of considering it a key contribution, since we had deemed it more of a "relative benefit to our approach". Following the reviewer's suggestion, we have further emphasized it by adding the following paragraph to Section 2: *FedVI's statelessness stems from its use of an "amortized posterior inference model." This concept shares similarities with how Variational Autoencoders (VAEs) perform inference. In VAEs, an encoder network maps input data to a latent space. FedVI takes a set of examples and infers the posterior over the local parameters,  effectively capturing the personalized features of each client's data. This dynamic encoding allows for continuous adaptation and personalization as new clients join and contribute their data.*

---

> > ### Author Response · Authors · 2024-04-25
> > **Reviewer Responses continue**
> >
> > **(c) I’m not convinced that this is actually doing VI (i.e. targeting the ELBO), as the expectation in (8) is now over the approximate posterior given the support set and the likelihood is only evaluated at a non overlapping query set. This feels closer to the neural process ML objective (e.g. Foong et al. (2020): Meta-learning stationary stochastic process prediction with convolutional neural processes, section 3.2.).**
> >
> > The reviewer's comment provided valuable insight regarding the potential deviation from standard VI and the similarity to the neural process ML objective. We acknowledge that splitting data into non-overlapping support and query sets could indeed compromise the validity of the ELBO bound. While we initially followed the approach in FedRecon we recognize the importance of adhering to proper VI principles. Therefore, we conducted additional experiments where we combined the support and query sets, effectively avoiding the data split. Encouragingly, these experiments yielded similar performance results, suggesting that the original approach did not significantly impact the overall findings.
> >
> > **How the posterior constructor model operates as a set function on the set of global features of the query set. Something that would help here is clearly defining the space in which the intermediate variables (e.g. $R^g_{k,q}$
> > ) exist.**
> >
> > We thank the reviewer for their question regarding the posterior constructor model's operation and the space of intermediate variables. We apologize for the confusion caused by our previous wording. To clarify, the posterior constructor model does not operate as a set function on the global features of the query set. Its input consists of the first $d_g$ elements (global features) of the embedding of each data sample in the support set. This distinction is important, as the model operates on a batch of individual data embeddings rather than sets of features.
> >
> > As you suggested, we have now explicitly defined the space of intermediate variables in Section 5. Specifically:
> >  *$R_{k,s}^g \in \mathbb{R}^{d_g}, R_{k,s}^l \in \mathbb{R}^{d_l}$ represent the global and local features, respectively, for the support set of client $k$.*
> >
> >  *$R_{k,q}^g \in \mathbb{R}^{d_g}, R_{k,q}^l \in \mathbb{R}^{d_l}$ represent the global and local features, respectively, for the query set of client $k$.*
> >
> > Here, $d$ is the embedding size, $d_g$ is  is the number of global features, and $d_l$ is the number of local features, with $d = d_l+d_g$.
> >
> > **3. I understand the notion of “stateless” to mean “no local parameters”. It would be great as to why this is beneficial in this case. Citations to use cases in the introduction would help, and clarity on why it’s an issue to store local parameters vs. computing them using a global model (aren’t these then stored locally anyway to obtain the local predictions?).**
> >
> > We agree with the reviewer's definition of statelessness, where clients do not retain local parameters. Our method aligns with this by using a latent set of local parameters that are marginalized during prediction, but not directly stored by clients. This stateless approach avoids the ``cold start" problem for new or infrequent clients, who might otherwise have poorly initialized local parameters. In contrast, stateful methods can create a participation gap, favoring clients with more training history. Our approach addresses this by treating local parameter estimation as an inference subproblem, enabling personalization without requiring clients to maintain a state.
> >
> > While stateful methods can offer advantages in scenarios with consistent client participation, our stateless approach aligns well with the practical realities of many real-world federated learning (FL) deployments. As highlighted in Table 1 of kairouz et al. 2021, statelessness is a defining characteristic of cross-device FL, emphasizing its importance in this context. Note that in practical cross device FL, clients participate in only a few rounds or less, leading to potentially stale or untrained local parameters that negate the benefits of statefulness. Our method is well-suited for such dynamic FL environments, where maintaining client state is challenging. Notably, most existing personalization attempts (pFedBayes, FedRep, APFL, clusteredFL) rely on stateful setups, whereas ours does not.
> >
> > **I’m also somewhat confused by “local parameters remain on clients” in section 3.1—isn’t this exactly what you’re trying to avoid by being stateless?**
> >
> > We agree with the reviewer that section 3.1 as written was confusing as to the treatment of "local parameters".  We have revised this section to read *"Global parameters update at the server end after each training round, while local parameters are deleted after each round."* in order to note that we do not maintain their state between rounds.

---

> > > ### Author Response · Authors · 2024-04-25
> > > **Review Responses continue**
> > >
> > > **4. There is no description of the baseline methods. Are the architectures used the same as the global model? At the bare minimum these should be included in the appendix.**
> > >
> > > We appreciate the reviewer's feedback regarding the baseline method descriptions. In the revised manuscript, we've added a paragraph as a detailed breakdown of the baseline architectures in Section 5.
> > >
> > > **5. “No assumptions are made about clients’ data generating distributions”. I’m not sure what you mean by this, as you’re using a very specific CNN architecture and categorical distribution which indicates that you are making assumptions.**
> > >
> > > We apologize for the confusion.  Our intent was to remind the reader that (as mentioned in line 5 of equation 1), that we have not assumed that the data generating distributions for each client are the same.  We of course do make assumptions about the support of the predictive and generative distributions, and the dimensionality of the event space.  We have clarified that statement to read *``We assume that clients' data generating distributions have the same support, but that they can be different from one another."* to be more specific about our assumptions.

---

> ### Comment · Reviewer_BQTq · 2024-05-01
>
> Many thanks for your detailed response to my concerns / questions. I welcome the suggested changes. However, I have a few questions remaining:
>
> ***Upper bound to the true risk***
>
> Just to clarify—equation 9 shows that equation 8 *plus a slack term* is an upper bound on the true risk, no?
>
> ***Set functions***
>
> “Its input consists of the first dg elements (global features) of the embedding of each data sample in the support set”
> I apologise, I had meant to say “set of global features of the support set”.  I believe this still constitutes a set function, as the support set is exchangeable in its elements and can be of arbitrary size. How are the global features of each datapoint in the support set combined?
>
> ***Local parameters***
>
> Regarding my initial confusion regarding “local parameters remain on clients”, is it not true that the local classifiers remain on clients? i.e. these are retained between rounds?
>
> ***Changes to Figures 2 and 3***
> One final comment I'd like to make is that I believe much of my confusion could be avoided by changing Figures 2 and 3 to include a much clearer, more accurate (detailed) diagrammatical illustration of the model.

---

> > ### Author Response · Authors · 2024-07-16
> >
> > We thank the reviewer for their thoughtful review and for acknowledging the suggested changes. Responses to the remaining questions are provided in the following:
> >
> > 1. Upper bound to the true risk: Yes, the reviewer is correct. Equation 9 establishes an upper bound on the true risk by adding a slack term to Equation 8.
> >
> > 2. Set functions: Apologies for the confusion. The global features are independently passed to the posterior reconstruction model and the outputs are subsequently averaged. We note that this is only one out of many ways in which one could combine the elements of each datum in the input. We tried a few alternatives (e.g. attention pooling) and found that they more or less performed equivalently.
> >
> > 3. Local parameters: We thank the reviewer for this clarifying question. You are correct that the term "local parameters" may have caused confusion. To be precise, our method does not retain the specific values of the local classifier parameters on each client between rounds. Instead, at each communication round: a) A new local posterior distribution is generated on each client using the posterior construction model. b) Fresh local classifier parameters are then sampled from this newly generated distribution. This is what we meant by the method being "stateless" – the clients do not store and update a fixed set of parameters over multiple rounds. Rather, they generate new parameters from a continuously evolving posterior distribution. We apologize for any confusion caused by our previous wording and hope this clarification addresses your question.
> >
> > 4. Changes to Figures 2 and 3: We appreciate the reviewer's feedback regarding Figures 2 and 3. We strive for clarity and accuracy in our visual representations, and we welcome specific suggestions on how to further enhance these figures. If the reviewer could kindly identify particular areas where more detail or clarity would be beneficial, we would be happy to incorporate those revisions to better illustrate the model's architecture and functionality.

---

### Review · Reviewer_QSyx · 2024-02-12

**Summary Of Contributions:**

This paper presents an algorithm for learning a variational approximation for local posteriors in federated deep learning task. The motivation for the work is that due to heterogeneity among the clients, learning a single global model might lead to poor generalization. Authors also consider a stateless setting, where we cannot assume that clients can retain the local model (nor the local model) at their device.

Instead of having a single global model, authors propose a hierarchical model for the data generating process over the clients. Using existing literature, authors are able to prove a PAC Bayes type of result. Their Corollary essentially says, that with probability $>1-\delta$, learning a VB approximation for the proposed model, reduces the the risk over the true data generating distribution (where the slack term of the risk is dependent on the $1/\delta$).

On the implementation side, authors learn three separate (global) models: (1) the embedding model, which maps images into a latent space, (2) a global prediction model, and (3) a posterior construction model that outputs parameters for clients variational posterior of the local predictive model. The scores from the models (2) and (3) are combined before computing the prediction loss. The posterior reconstruction model (2), allows the stateless setting. I think the posterior reconstruction model works a lot like the encoding part of a VAE, which maps each data point to the particular posterior means and stds. In the setting of this paper, the posterior parameters are obtained for a batch of client's data.

The performance of the proposed method is compared against various FL approaches on two data sets FEMNIST and CIFAR-100. On both of these data sets, the proposed algorithm beats the other methods with somewhat significant margin.

**Audience:**

Yes

**Broader Impact Concerns:**

I see no immediate unaddressed broader impact concerns in this work.

**Claims And Evidence:**

No

**Requested Changes:**

**Major**
- Further experiments on whether $\tau=0$ for the FEMNIST leads to significantly larger generalization gap than $\tau > 0$.
- Clarify how the merging of local and global predictions is done

**Minor**
- What $\tau$ value was used for the results in Table 1 and Fig. 4?
- Page 4: "the best approximation", in general this is not true since the quality of the approximation is completely dependent on the chosen surrogate distribution.
- Page 4: The formula in the end: I guess this is actually negative ELBO?
- Corollary 1: Please clarify what is the "slight generalization" over the Germain et al. 2016. To me, the bound in equation (9) looks very much the same as the bound in the prior work.
- About the local parameters. On the 7th point of the listing in page 7, you say that "Both local and global parameters get updated through back propagation ...". So I guess as local parameters you mean the $\mu_k$ and $\sigma_k$, and not the $\beta$ (which is titled as local parameters on eq (18))?
- When you say that "The local and global predictions are merged to get the predictions", how is the merging done? The samples fed to both classifiers are the same (just different parts of the feature vector) right?
- I guess Figure 4 is not referred anywhere in the main text?
- About Figure 5: If I have understood correctly, the $\tau$ parameter controls the level of local regularization through scaling the KL divergence between the variational posterior and the prior for $\beta_k$ parameter. Could you clarify, would the $\tau=1$ correspond to the actual ELBO objective? Or is there some scaling issue between the data and the prior that might lead to $\tau \neq 1$ being the correct scale? If not, then it would be interesting to have more discussion why the test performance gap seems to somewhat high for the $\tau=1$ in the FEMNIST case, as I would imagine it should lead to better generalization that the $\tau < 1$.

- Appendix C: in the first line of eq. (24), it seems like you are writing $\ell(Y \mid X) = E_{q(\theta, \beta^c \mid X, Y)} [\ell(Y \mid X, \theta, \beta^c)]$. I assume that the $\ell(Y \mid X)$ and $\ell(Y \mid X, \theta, \beta^c)$ are the evidence and likelihood. Is this really true for variational $q$? Wouldn't you have that $\ell(Y \mid X) = E_{\pi(\theta, \beta^c)} [\ell(Y \mid X, \theta, \beta^c)]$? Or if $q$ is not the variational approximation, can you please clarify what it is?
- Appendix C: what is the "||" after the 2nd $\leq$ (close to equation 24)? Is it supposed to be KL?
- Appendix C: I believe the last expectation in the Donsker Varadhan inequality should have a log around it (talking about the expression after "After that we use the Donsker-Varadhan inequality which says").

**Very minor**
- Page 14: "Morkov's inequality"
- Some of the references look a bit odd. There are a lot of uncapitalized letters, as well as missing venues.
- Possibly relevant cite: Han Guo, Philip Greengard, Hongyi Wang, Andrew Gelman, Yoon Kim, Eric P. Xing: Federated Learning as Variational Inference: A Scalable Expectation Propagation Approach. ICLR 2023

**Strengths And Weaknesses:**

# Strengths:
I believe the motivation, both the personalization and the statelessness, for the work is well justified and something that would definitely be of interest for the TMLR audience. Also the connection between you generalization bound in eq. (9) and the ELBO itself is an interesting realization. The empirical evaluation seems to suggest that the proposed model outperforms the existing approach, even with some margin.

# Weaknesses:
### Writing
The writing of the paper needs to improve quite significantly. I do not quite understand for example how the scores computed by the local and global model are "merged" before updating the model. This would be a crucial bit of information for understanding how the local predictions can help in personalization.

### Experiments:
So do I understand correctly, that if you set $\tau=0$ in your experiments, then you don't get any regularization from the local priors? If so, I guess the proposed setting wouldn't differ from having a single global model (assuming $\tau=0$)? Now, looking at Fig. 5a, while there are some points where the generalization gap is smaller than with $\tau=0$, it seems to be quite hard to predict how different $\tau > 0$ values work. Would it be possible to somehow repeat the inference multiple times to get less noisy estimates on how the test errors behave as a function of $\tau$? I'm mainly worried that since the FEMNIST is most likely the data set that has some strong heterogeneity among the clients, and if we cannot clearly witness the regularization there, then it might be hard to say if the regularization actually has some statistically significant effect.

---

> ### Author Response · Authors · 2024-04-25
> **Review Responses**
>
> The reviewer’s insightful comments have significantly strengthened our paper, and we are grateful for their feedback. In the following we provided detailed responses to the reviewer comments.
>
> **1. Further experiments on whether $\tau=0$ for the FEMNIST leads to significantly larger generalization gap than $\tau>0$.**
>
> We conducted additional experiments, running the FEMNIST scenario five times with different seed values $(0, 10, 20, 30, 40)$. We then calculated the average and variance of both hold-out and participating test accuracy across these runs. The results, presented in the revised figure (please find the figure in the response to this question in the uploaded file), provide a smoother visualization of the trend. As the figure demonstrates, we still observe a noticeably larger generalization gap for $\tau=0$ compare to $\tau>0$. This supports our initial findings and further emphasizes the importance of the KL divergence term in promoting generalization, even with increased experimental rigor.
>
> We appreciate the reviewer's insightful suggestion, which has helped us strengthen the robustness of our results and provide a clearer understanding of the impact of $\tau$ on model performance.
>
> **2. Clarify how the merging of local and global predictions is done.**
>
> We thank the reviewer for this suggestion.  While there are multiple ways that one could merge the predictions, we found that the simplest way was to add them together.  This treats the local predictions as modifications to the global predictions in Logit space. To improve clarity for the reviewer, we have added an explanation as a footnote on page 8.
>
> **3. What $\tau$ value was used for the results in Table 1 and Fig. 4?**
>
> We used $\tau=10^{-9}$ for FEMNIST, and $\tau=10^{-3}$ for CIFAR-100 in both table 1 and Figure 4, since each performed best in our experiments.  In order to make this clearer, we have mentioned these values in the captions of Table 1 and Figure 4 in the revised manuscript.
>
> **4. Page 4: "the best approximation", in general this is not true since the quality of the approximation is completely dependent on the chosen surrogate distribution.**
>
> In order to address the reviewer's concern, we have changed this statement to read *"This minimization process provides the best approximation for the intractable posterior distribution, under the chosen family of surrogates"*.  This highlights that, while this minimization may not yield the best approximation to the true posterior over all distributions, it does produce the closest surrogate to the true posterior (in terms of the KL divergence between them), out of the chosen family of surrogates.
>
> **5. Page 4: The formula in the end: I guess this is actually negative ELBO?**
>
> We thank the reviewer for their comment. We have revised the paper as suggested.
>
> **6. Corollary 1: Please clarify what is the "slight generalization" over the Germain et al. 2016. To me, the bound in equation (9) looks very much the same as the bound in the prior work.**
>
> We appreciate the reviewer's question regarding Corollary 1. We acknowledge that the bound in Equation (9) appears similar to the bound in (Germain et al. 2016). However, our work offers a slight generalization in the following way:
>
> 1. Non-IID Data: While Theorem 3 in (Germain et al. 2016) explicitly assumes IID empirical data samples, our corollary extends the applicability to non-IID cases. This is particularly relevant in federated learning scenarios where data distributions can vary across clients.
>
> 2. Proof Adaptation: In the proof of our corollary, we demonstrate that the core argument in (Germain et al. 2016) does not strictly require the IID assumption for empirical data. This allows us to derive a similar bound even when data samples originate from different distributions.
>
> We believe this generalization, though seemingly subtle, is an important contribution as it broadens the theoretical foundations of PAC-Bayes bounds in the context of federated learning and other non-IID settings.

---

> > ### Author Response · Authors · 2024-04-25
> > **Review Responses continue**
> >
> > **7. About the local parameters. On the 7th point of the listing in page 7, you say that "Both local and global parameters get updated through back propagation ...". So I guess as local parameters you mean the $\mu_k$ and $\sigma_k$, and not the $\beta$ (which is titled as local parameters on eq (18))?**
> >
> > We appreciate the reviewer's attention to detail regarding the terminology of local parameters. The reviewer is correct that in Equation (18), $\beta_k$ is referred to as the local parameters.  Our method constructs a distribution over these parameters, using ($\mu_k$ and $\sigma_k$) and then samples $\beta_k$ from that distribution.  However, all of these are inferred from the global reconstruction model.  It is the global parameters of that model which are updated through backpropagation.  We have changed the wording of the seventh point on {page 8} in the revised manuscript to make this explicit.  Note that by updating the global parameters, we are implicitly updating the local parameters (since we use the reconstruction model to infer them).
> >
> > **8. When you say that "The local and global predictions are merged to get the predictions", how is the merging done?**
> >
> > We thank the reviewer for this question.  While there are multiple ways that one could merge the predictions, we found that the simplest way was to add them together.  This treats the local predictions as modifications to the global predictions in Logit space. To improve clarity for the reviewer, we have added an explanation as a footnote on page 8.
> >
> > **The samples fed to both classifiers are the same (just different parts of the feature vector) right?**
> >
> > Yes, this is correct. The same samples are fed to both classifiers, but different portions of the feature vector are used. As detailed in the "Data partitioning" section (pages 7-8), we split the feature vector into global and local components. In our experiments, we found that using a larger proportion of global features (80%) compared to local features (20%) yielded the best performance. Specifically, when the embedding model's last layer has a dimension of $d = 128$, the first 102 features are considered global, and the remaining 26 are local.
> >
> > **9. I guess Figure 4 is not referred anywhere in the main text?**
> >
> > We thank the reviewer for bringing this to our attention. While we initially provided a brief explanation in the caption, we understand that this was insufficient. To address this, we have added the following explanation to the ``Evaluation Results and Discussion" section: *Figure 4 illustrates the non-participating test accuracy on FEMNIST ( $\tau = 10^{-9}$ ) and CIFAR-100 ($\tau = 10^{-3}$) over $1500$ rounds of training, providing a visual representation of the results reported in Table 1.*
> >
> > **10. About Figure 5: If I have understood correctly, the $\tau$ parameter controls the level of local regularization through scaling the KL divergence between the variational posterior and the prior for $\beta$ parameter. Could you clarify, would the $\tau = 0$ correspond to the actual ELBO objective? Or is there some scaling issue between the data and the prior that might lead to $\tau \neq 0$ being the correct scale? If not, then it would be interesting to have more discussion why the test performance gap seems to somewhat high for the $\tau = 0$ in the FEMNIST case, as I would imagine it should lead to better generalization that the
> > $\tau < 1$.**
> >
> > We appreciate the reviewer's insightful question regarding the role of the $\tau$ parameter and its relationship to the ELBO objective. That is correct that $\tau$ controls the level of local regularization by scaling the KL divergence between the variational posterior and the prior for the $\beta$ parameter.
> >
> > Our objective function in Equation (8) is indeed a generalization of the ELBO, where setting $\tau$ = $\gamma$ = 1 recovers the standard ELBO. Setting $\tau=0$ effectively removes the KL divergence term, resulting in maximum likelihood estimation (MLE).
> >
> > This distinction explains the higher generalization gap observed for $\tau = 0$ in both FEMNIST and CIFAR-100 experiments (Figure 5). While MLE can sometimes lead to overfitting, the KL divergence term in the ELBO acts as a regularizer, promoting better generalization. Therefore, values of $\tau>0$, which maintain the KL divergence term, generally exhibit superior performance and result is smaller generalization gaps.
> >
> > We hope this explanation clarifies the behavior of $\tau$ and its impact on generalization.

---

> > > ### Author Response · Authors · 2024-04-26
> > > **Review Responses continue**
> > >
> > > **11. Appendix C: in the first line of eq. (24), it seems like you are writing $\ell(Y|X) = E_{q(\theta,\beta^c|X,Y)}[\ell(Y|X,\theta, \beta^c)]$. I assume that the $\ell(Y|X)$ and $\ell(Y|X, \theta,\beta^c)$ are the evidence and likelihood. Is this really true for variational $q$? Wouldn't you have that $\ell(Y|X) = E_{\pi(\theta,\beta^c)}[\ell(Y|X,\theta, \beta^c)]$? Or if $q$ is not the variational approximation, can you please clarify what it is?**
> > >
> > > We appreciate the reviewer's insightful question regarding Equation (24) in Appendix C. Throughout this paper $\ell(Y|X)$ stands for marginalized likelihood over the surrogate posterior so $\ell(Y|X) = E_{q(\theta,\beta^c|X,Y)}[\ell(Y|X,\theta, \beta^c)]$ is correct. To clarify our notation, we use $\nu(X)$ to denote the prior data generating distribution. Therefore, we would show the evidence by $\nu(Y|X) = E_{\pi(\theta,\beta^c)}[\ell(Y|X,\theta, \beta^c)]$.
> > >
> > > **12. Appendix C: what is the $||$ after the 2nd $\leq$ (close to equation 24)? Is it supposed to be KL?**
> > >
> > > We thank the reviewer for spotting the typo in Appendix C. The symbol "$||$" after the second "$\leq$" should indeed be "$D_{kl}$". We have corrected this in the updated manuscript.
> > >
> > > **13. Appendix C: I believe the last expectation in the Donsker Varadhan inequality should have a log around it (talking about the expression after "After that we use the Donsker-Varadhan inequality which says").**
> > >
> > > We appreciate the reviewer's careful attention to detail. The typo has been corrected in the revised version.
> > >
> > > **14. Some of the references look a bit odd. There are a lot of uncapitalized letters, as well as missing venues.**
> > >
> > > We thank the reviewer for pointing out the reference formatting inconsistencies. We apologize for this oversight and have carefully reviewed our bib file to identify and correct the issues. In the updated manuscript, all references now include venues, ensuring clarity and adherence to formatting guidelines.
> > >
> > > **15. Possibly relevant cite: Han Guo, Philip Greengard, Hongyi Wang, Andrew Gelman, Yoon Kim, Eric P. Xing: Federated Learning as Variational Inference: A Scalable Expectation Propagation Approach. ICLR 2023**
> > >
> > > We are grateful for the reviewer for suggesting a relevant reference. We have incorporated this reference in Section 5 and added FedEP to the baseline comparisons in Table 1, strengthening the evaluation of our method.

---

> > > > ### Comment · Reviewer_QSyx · 2024-05-06
> > > > **After rebuttal**
> > > >
> > > > Thanks for your thorough response!
> > > >
> > > > I'm happy with the updated FEMNIST experiment, and it seems to provide further evidence on the benefits of the regularization with $\tau > 0$. However, I would suggest authors to carefully discuss these results in the paper. From the updated plot, it seems that there are some $\tau > 0$ values that lead to almost the same generalization gap as $\tau = 0$. Moreover, there is no clear trend w.r.t the $\tau$, so at least according to these results it does not seem that more regularization necessarily leads into better generalization.
> > > >
> > > > Looking at the results again, it seems that the held-out set constantly achieves better predictive performance than the participating clients. Do you have an idea why this is? Even for the $\tau=0$ (which  boils down to MLE), the prediction for the held-out set seems to be better.
> > > >
> > > > Finally, all my minor comments have been sufficiently answered. One minor suggestion regarding comment 11 (in your above numbering): please make sure that this notation is discussed in the paper (talking about the $\ell(Y \mid X)$).

---

> > > > > ### Author Response · Authors · 2024-07-16
> > > > >
> > > > > 1. We appreciate the reviewer's positive feedback on the updated FEMNIST experiment and the recognition of its contribution to understanding the effects of regularization with $\tau>0$.  As suggested, we have added Appendix D, which offers a detailed discussion of the experiment and its findings.
> > > > >
> > > > > 2. We appreciate the reviewer's keen observation regarding the performance difference between the held-out and participating clients. The initial experiment involved holding out only $20$ clients ($0.58$% of the population) to ensure a fair comparison with other baselines.  This small held-out set, might not fully represent the overall data distribution due to the inherent stochasticity in the remaining 3380 participating clients. To investigate this further, we conducted additional experiments on FEMNIST for $\tau = 10^{-9}$ with varying sizes of held-out sets ($170$, $340$, $680$, $850$, and $1700$ clients, representing $5$%, $10$%, $20$%, $25$%, and $50$% of the population). As shown in Figure 7 in Appendix D, these results indicate that increasing the size of the held-out set leads to an increase in participation gap, with the participating clients eventually outperforming the nonparticipating ones. This suggests that the initial observation was indeed influenced by the small size of the held-out set and its potential deviation from the overall data distribution.
> > > > >
> > > > > 3. We appreciate the reviewer's confirmation that all minor comments have been addressed. The notation $\ell(Y \mid X)$ is now clarified in a footnote in Appendix C, as suggested.

---

### Review · Reviewer_WAV2 · 2024-04-09

**Summary Of Contributions:**

This paper introduces Federated Variational Inference (FedVI), a novel algorithm designed specifically to enhance personalization and generalization in federated learning (FL). The authors claim that FedVI more effectively handles data heterogeneity. The FL process is formulated using a hierarchical generative model and formalized using Bayesian Inference. Variational Inference is then used to enhance learning efficiency. Theoretical analyses are provided to justify the performance of FedVI. Empirical studies demonstrate that FedVI outperforms state-of-the-art methods.

**Audience:**

Yes

**Broader Impact Concerns:**

I don't find any Broader Impact concerns in the current manuscript.

**Claims And Evidence:**

Yes

**Requested Changes:**

As briefly mentioned in the "weakness" section, it would be essential for the author to make the following modifications:
- Adding experiments on the extra computation and communication overheads of FedVI. And study the end-to-end wall clock time comparisons between FedVI and other baseline methods.
- The experimental scales are too small. It would be essential to demonstrate the effectiveness of FedVI on larger-scale datasets, e.g., ImageNet.
- Add comparisons between FedVI and baseline methods like [1-2].

[1] https://arxiv.org/abs/2010.05273
[2] https://arxiv.org/abs/2302.04228

**Strengths And Weaknesses:**

Strengths:
- The paper is well-written and well-motivated in general. Improving the effectiveness of FL is a promising research direction.
- The presented idea behind FedVI is easy to follow.
- Both theoretical and experimental results are presented to justify the method.

Weaknesses:
- It is not clear about the extra computation and communication incurred by FedVI. By looking at the major algorithm box, FedVI seems to be much more complex compared to FedAvg.
- The extra computation and communication costs have not been explicitly studied in the experiments.
- The experimental scales are too small. It would be essential to demonstrate the effectiveness of FedVI on larger-scale datasets, e.g., ImageNet.
- Some similar baseline methods are missing, e.g., [1-2].

[1] https://arxiv.org/abs/2010.05273
[2] https://arxiv.org/abs/2302.04228

---

> ### Author Response · Authors · 2024-04-26
> **Review Responses**
>
> We would like to thank the reviewer for the helpful comments, which have greatly improved the paper. In what follows we provided point by point responses to the reviewer's comments.
>
> **1. Adding experiments on the extra computation and communication overheads of FedVI. And study the end-to-end wall clock time comparisons between FedVI and other baseline methods.**
>
> We found the reviewer's suggestion to be particularly valuable to analyze the computation and communication overheads of FedVI compared to other baseline methods.
>
> To assess the complexity of FedVI algorithm as compared to FedAvg, we conducted experiments where we removed the posterior reconstruction and local classifier components from FedVI, effectively turning it into FedAvg. We then compared the runtime of both algorithms on the FEMNIST dataset for 200 rounds using an NVIDIA A100 GPU (all the other hyperparameters are similar to what is reported in the main manuscript).
>
> The results presented in train-runtime and eval-runtime figures (please find these figures in the uploaded file in response to this question) demonstrate comparable training and evaluation runtimes for both FedAvg and FedVI. FedVI achieves an average training runtime of $71.35$ seconds per round and evaluation runtime of $18.96$ seconds per round. In comparison, FedAvg training requires $71.74$ seconds per round and evaluation takes $18.39$ seconds per round on average. While FedVI incorporates additional components compared to FedAvg, our experiments demonstrate that these complexities do not significantly impact runtime. This is because the added components do not significantly contribute to the computational overhead, as the primary bottleneck is the embedding model. Similarly, although FedVI requires communicating slightly more global parameters (posterior constructor parameters), this does not notably affect runtime. These results underscore that FedVI's personalization benefits come at a minimal cost in terms of computational and communication efficiency.
>
> We acknowledge the reviewer's request for comparisons with other baselines. While a direct runtime comparison is challenging without re-implementing those methods, we can offer some insights. Since the posterior constructor model and local classifier do not represent computational bottlenecks for FedVI, its runtime is expected to be similar to or potentially even less than that of other baseline methods. This is because those baselines typically involve personalization components that may introduce additional computational or communication overhead compared to FedAvg.
>
> We hope this analysis provides valuable insights into FedVI's computational and communication efficiency.
>
> **2. The experimental scales are too small. It would be essential to demonstrate the effectiveness of FedVI on larger-scale datasets, e.g., ImageNet.**
>
> We thank the reviewer for suggesting the evaluation of FedVI on ImageNet and other large-scale datasets. While we acknowledge the general value of such experiments, we believe that applying FedVI to ImageNet might not provide the desired insights for the following reasons:
>
> 1. Federated Nature: ImageNet is not inherently structured for federated learning (FL). Adapting it to an FL setting would require simulating an artificial client-based distribution, which might not accurately reflect real-world FL scenarios. This could potentially distort the evaluation of our personalized method, as its effectiveness relies on the natural heterogeneity and distribution shifts present in genuine FL data. Therefore, we believe that evaluating on datasets specifically designed for FL or those with inherent client-based structures (such as FEMNIST) would provide a more reliable demonstration of our method's capabilities.
>
> 2. Baseline Comparisons: Our primary focus here lies in personalized federated learning, where existing baselines are rarely evaluated on datasets of that scale. This makes direct comparisons with other FL methods on ImageNet challenging. We believe the chosen datasets, specifically FEMNIST, effectively demonstrate FedVI's effectiveness in a personalized FL setting and exhibit inherent heterogeneity and client-based distribution shifts, which represent the core challenges that FedVI addresses.
>
> **3. Add comparisons between FedVI and the suggested baseline methods.**
>
> Following the reviewer's helpful suggestion, we have integrated the recommended references on FedEP and FedPA into Section 5. Additionally, we have included these methods as baselines in Table 1 to provide a more comprehensive evaluation.

---

### Author Response · Authors · 2024-04-25

We sincerely thank all reviewers for their time, effort, and valuable feedback, which has significantly improved our paper's quality. We have uploaded our response to OpenReview, including a detailed address of each comment, results from requested experiments, and an updated manuscript with changes highlighted in blue. We hope we have addressed all concerns and welcome further discussion if needed.

---

### Author Response · Authors · 2024-07-15

We are deeply grateful to the reviewers and action editor for their positive decision and invaluable feedback throughout this process. The additional comments on our response were particularly helpful in refining the final version of our paper.

---

### Decision · Action_Editor_tW7Q · 2024-06-12

**Recommendation:** Accept with minor revision

**Comment:**

The contribution and results of the paper are in principle sufficient for TMLR. I am nevertheless recommending acceptance subject to minor revision through addressing the following points:
1. Notation: the paper is sloppy by using $D_{KL}$ as a shorthand for $E_q \log(q/p)$ even when $q$ and $p$ are not defined over the same space (see e.g. Eqs. 6 and 8 + Appendix). This can be very confusing to readers because this does not follow usual rules of KL divergence, such as non-negativity. In order for the paper to be acceptable, this would need to be clarified, ideally by reserving the notation $D_{KL}$ just for the KL divergence.
2. Related to the comment 11 of reviewer QSyx, the "True risk" term in Eq. (9) does not appear to be the true risk because it is an expectation over $q$, not $p$. This would need to be clarified.
3. As requested by the reviewers, please expand the discussion of the FEMNIST results and their significance.
4. Your response included a new figure (Fig. 1 in the response) that does not seem to be in the revised paper. Please consider adding it.
5. The reviewers raised several followup comments in the discussion after the author response. Please review and respond to these.

Typo: Appendix C refers to "Morkov's" inequality.

**Audience:**

The paper makes interesting contributions for variational inference in memoryless federated learning, which is typical cross-device federated learning. All reviewers agree that there would be individuals in TMLR's audience interested in the findings of the paper.

**Claims And Evidence:**

All reviewers find that the claims are supported by sufficient evidence, although they raise concerns about the presentation.

I share these concerns, and feel the paper would still need a revision before being acceptable. Some more detailed comments below.

---

> ### Author Response · Authors · 2024-07-15
>
> 1. We thank the action editor for bringing this to our attention. We acknowledge the potential for confusion and have revised Equations 6 and 8 in the main text, along with Equation 22 in Appendix A, to reserve the notation $D_{KL}$ exclusively for the KL divergence and use alternative notation where appropriate.
>
> 2. We thank the action editor for pointing out this potential source of confusion. In full generality the term ``True Risk" refers to the population risk which is differentiated from the empirical risk which also appears in Equation 9 and is defined for a sample from $\mathcal{D}$ rather than an expectation over $\mathcal{D}$. Theorem 3 of Germain et al. and Corollary 1 of our paper showed that the bounds hold for any choice of distribution over the hypothesis set. This includes the true posterior $p$, the surrogate posterior $q$, or any other valid distribution. Since we are optimizing the surrogate, it make sense to use it in Equation 9. We are happy to discuss this further if the action editor has any additional questions or concerns.
>
> 3. We have addressed the reviewers' request by expanding the discussion of the FEMNIST results in Appendix D.
>
> 4. We have added new figures for FEMNIST dataset to the revised paper, as suggested. This figure is now included as Figure 6 in Appendix D.
>
> 5. We have carefully reviewed and responded to all of the reviewers' follow-up comments in the discussion thread.
>
> 6. We thank the action editor for pointing out this typographical error. We have corrected the spelling to ``Markov's inequality" in Appendix C.